# Deep learning-based image analysis predicts PD-L1 status from H&E-stained histopathology images in breast cancer

Gil Shamai [1,7] ✉, Amir Livne[1,7], António Polónia [2], Edmond Sabo[3], Alexandra Cretu[3], Gil Bar-Sela[4,5] & Ron Kimmel[1,6]

Programmed death ligand-1 (PD-L1) has been recently adopted for breast cancer as a predictive biomarker for immunotherapies. The cost, time, and variability of PD-L1 quantification by immunohistochemistry (IHC) are a challenge. In contrast, hematoxylin and eosin (H&E) is a robust staining used routinely for cancer diagnosis. Here, we show that PD-L1 expression can be predicted from H&E-stained images by employing state-of-the-art deep learning techniques. With the help of two expert pathologists and a designed annotation software, we construct a dataset to assess the feasibility of PD-L1 prediction from H&E in breast cancer. In a cohort of 3,376 patients, our system predicts the PD-L1 status in a high area under the curve (AUC) of 0.91 – 0.93. Our system is validated on two external datasets, including an independent clinical trial cohort, showing consistent prediction performance. Furthermore, the proposed system predicts which cases are prone to pathologists missinterpretation, showing it can serve as a decision support and quality assurance system in clinical practice.

Breast cancer became the leading cause of death in women ages 20 to 59 and the most diagnosed cancer as of 2021, accounting for 12% of all new annual cancer cases worldwide[1]. Immunotherapy for programmed death 1 (PD-1) and programmed death ligand-1 (PD-L1) is one of the promising recently developed treatments for several types of cancer. Such treatments trigger the immune system to fight the cancer by blocking the association between PD-L1 and PD-1 checkpoints that suppresses the immune system. In lung cancer, PD-L1/PD-1 inhibitors immunotherapy was recently found to have better overall response and overall survival for patients with tumor expression of over 50% PD-L1 in tumor cells compared to standard chemotherapy[2]. Following its success in lung cancer, as well as in other types of cancer, PD-L1 has recently gained attention as a predictive biomarker for immunotherapy response in triple negative breast cancer and other subtypes of breast cancer[3–5]. For example, the IMpassion130 study showed that adding immunotherapy to

chemotherapy had no benefit for overall survival time in triple negative breast cancer patients. When considering only the group of PD-L1 positive patients, however, addition of immunotherapy has significantly improved the survival time[6]. Based upon this study, the Food and Drug Administration (FDA) has approved the assessment of PD-L1 by Ventana SP142-stained immunohistochemistry (IHC) in triple negative breast cancer for selecting patients to receive immunotherapy[7].

PD-L1 staining by IHC, which is the current conventional approach for PD-L1 assessment, is costly, time consuming, may exhaust the tissue, and is even inaccessible in some countries. Its interpretation is nontrivial, requires special expertize, and is more than often inconsistent. More than one staining method for PD-L1 exists, and several studies have shown that quantification of PD-L1 expression may significantly change, depending on the staining method and antibody used[8,9]. Other studies have shown low rates of repeatability for PD-L1

[1]Department of Computer Science, Technion, Haifa, Israel. [2]Department of Pathology, Ipatimup, Porto, Portugal. [3]Department of Pathology, Carmel Medical Center, Haifa, Israel. [4]Department of Oncology, Haemek Medical Center, Afula, Israel. [5]Faculty of Medicine, Technion, Haifa, Israel. [6]Department of Electrical and Computer Engineering, Technion, Haifa, Israel. [7]These authors contributed equally: Gil shamai, Amir Livne. ✉e-mail: gil.shamai@gmail.com

assessment by certified pathologists, even within each single staining method[10–12].

Hematoxylin and eosin (H&E) is the basic staining that is routinely done for every biopsy today, and allows visual examination of the tissue and cells. In contrast to IHC, H&E is robust, reliable, efficient, and cheap, and does not depend on the choice of antibodies. Based on H&E, pathologists detect cancer and diagnose its subtype and grade. Nevertheless, visual examination of H&E by pathologists is limited and is not used for predicting the expression of PD-L1 or other biomarkers.

Convolutional neural networks (CNNs) are a class of machine learning methods that are optimized for image analysis, and currently provide state-of-the-art performance for various image classification tasks. In computational pathology, image analysis by CNNs has shown comparable performance to pathologists in various tasks, such as IHC-based PD-L1 assessment and H&E-based tumor and grade classification, and has been recently approved by the FDA for quality assurance and decision support for tumor detection in clinical practice[13–18].

It has been shown that machine learning can reveal information in H&E images unseen by the human eye, and, therefore, unexploited in the pathology setting[19,20]. Shamai et al.[21] have recently demonstrated, for the first time, that molecular biomarker expression can be predicted by machine learning from H&E tissue microarray (TMA) images in breast cancer, without immunohistochemistry - a prediction that is yet beyond human interpretation and ability to reproduce. Moreover, they showed that Estrogen receptor (ER) status could be predicted with accuracy comparable to inter-pathologist quantification by IHC. More recent studies supported this evidence by using CNNs for predicting ER, Progesterone receptor (PR), and human epidermal growth factor receptor 2 (ERBB2) from H&E-stained from TMAs and whole slide images (WSI) in breast cancer[22–25], as well as other biomarkers in other types of cancer[26,27]. Nevertheless, no evidence has yet been found for the benefit of H&E-analysis for prediction of PD-L1 expression in breast cancer.

Here we show that PD-L1 expression can be predicted by CNN-based analysis of H&E images. Since PD-L1 has been recognized as an important biomarker in breast cancer only recently, it is not yet a part of the routine clinical practice, and it is thus challenging to construct large datasets of H&E images coupled with PD-L1 expression. In our study, we exploited a large tissue microarray repository containing H&E-stained images and multiple corresponding stains for various biomarkers, including IHC for PD-L1. An expert pathologist annotated the samples in the dataset for PD-L1 expression. The dataset we constructed allowed us to train and test a CNN for the prediction of PD-L1 expression from H&E images for the first time.

## Results

### PD-L1 in the BCCA and MA31 cohorts

The study was based on breast cancer tissue samples and clinicopathological data of 5596 patients with 26,763 TMA images from two independent cohorts: The British Columbia Cancer Agency (BCCA) and the MA31 (Table 1). The BCCA cohort is composed of 4,944 women with newly diagnosed invasive breast cancer in British Columbia, whose tumor specimens were processed by a central laboratory at Vancouver General Hospital between 1986 and 1992. Each woman had three H&E-stained TMA cores, one IHC-stained TMA for PD-L1, and one for PD-1.

The MA31 cohort is a clinical trial of the Canadian Cancer Trials Group, conducted from January 17, 2008, through December 1, 2011, and was designed to evaluate the prognostic and predictive biomarker utility of pretreatment serum PD-L1 levels. This cohort consists of 652 recruited patients with ERBB2-positive metastatic breast cancer from 21 countries. Each woman had between 1 to 4 H&E-stained images, and one PD-L1-stained image corresponding to each H&E image (Table 1).

An expert pathologist annotated the entire data, consisting of both BCCA and MA31 cohorts, for PD-L1 positive or negative status, by

going through all available H&E and PD-L1 TMA images (Fig. 1a). To make the annotation process easier, we designed a computer-aided application which enabled the pathologist to go through the patient images, alternate between TMA images of each patient, and determine their expression status using keyboard shortcuts and a place for typing comments (see "Methods"). Following the annotation process, part of the patients from both cohorts were excluded from the analysis due to one of the following reasons (Table 1): Missing TMA images, TMAs with no tissue or no tumor, deficient or non-specific staining, or images out of focus. The rest of the patients were classified as either negative or positive for PD-L1 status.

### Training and testing the system

To set up the data for training and inference, we pre-processed the H&E-stained TMA images. All images were cropped and resized from a resolution of $1440 \times 2560$ to $512 \times 512$ pixels, and data augmentation was performed to help the model deal with variability in staining methods and other differences between the cohorts (See "Methods"). The BCCA patients were randomly divided at the patient level to training (2516, 74.5%) and test (860, 25.5%) sets (Table 1).

The training set was further randomly divided at the patient level to five train, and validation folds and the system was then trained and

**Table 1 | Patients and TMAs included and excluded in each data group**

| Cohort | BCCA | | | | MA31 | |
|---|---|---|---|---|---|---|
| Biomarker | PD-L1 | | PD-1 | | PD-L1 | |
| Total patients | n | % | n | % | n | % |
| Total | 4944 | 100.0% | 4944 | 100.0% | 652 | 100.0% |
| Excluded from analysis | 1568 | 31.7% | 1449 | 29.3% | 377 | 57.8% |
| Included in analysis | 3376 | 68.3% | 3495 | 70.7% | 275 | 42.2% |
| Patients excluded from analysis | n | % | n | % | n | % |
| Total | 1568 | 100.0% | 1449 | 100.0% | 377 | 100.0% |
| No TMAs | 168 | 10.7% | 0 | 0.0% | 229 | 60.7% |
| No tissue | 969 | 61.8% | 1189 | 82.1% | 118 | 31.3% |
| No tumor | 176 | 11.2% | 188 | 13.0% | 25 | 6.6% |
| Deficient staining | 224 | 14.3% | 61 | 4.2% | 5 | 1.3% |
| Out of focus | 31 | 2.0% | 11 | 0.8% | 0 | 0.0% |
| Patients included in analysis | n | % | n | % | n | % |
| Total | 3376 | 100.0% | 3495 | 100.0% | 275 | 100.0% |
| Negative | 2819 | 83.5% | 3388 | 96.9% | 252 | 91.6% |
| Positive | 557 | 16.5% | 107 | 3.1% | 23 | 8.4% |
| Training set | 2516 | 74.5% | 2618 | 74.9% | 0 | 0.0% |
| Test set | 860 | 25.5% | 877 | 25.1% | 275 | 100.0% |
| H&E TMAs included in analysis | n | % | n | % | n | % |
| Total | 10,128 | 100.0% | 10,485 | 100.0% | 515 | 100.0% |
| Negative | 8457 | 83.5% | 10164 | 96.9% | 482 | 93.6% |
| Positive | 1671 | 16.5% | 321 | 3.1% | 33 | 6.4% |
| Training set | 7548 | 74.5% | 7854 | 74.9% | 0 | 0.0% |
| Test set | 2580 | 25.5% | 2631 | 25.1% | 515 | 100.0% |
| IHC TMAs included in analysis | n | % | n | % | n | % |
| Total | 3376 | 100.0% | 3495 | 100.0% | 515 | 100.0% |
| Negative | 2819 | 83.5% | 3388 | 96.9% | 482 | 93.6% |
| Positive | 557 | 16.5% | 107 | 3.1% | 33 | 6.4% |
| Training set | 2516 | 74.5% | 2618 | 74.9% | 0 | 0.0% |
| Test set | 860 | 25.5% | 877 | 25.1% | 515 | 100.0% |

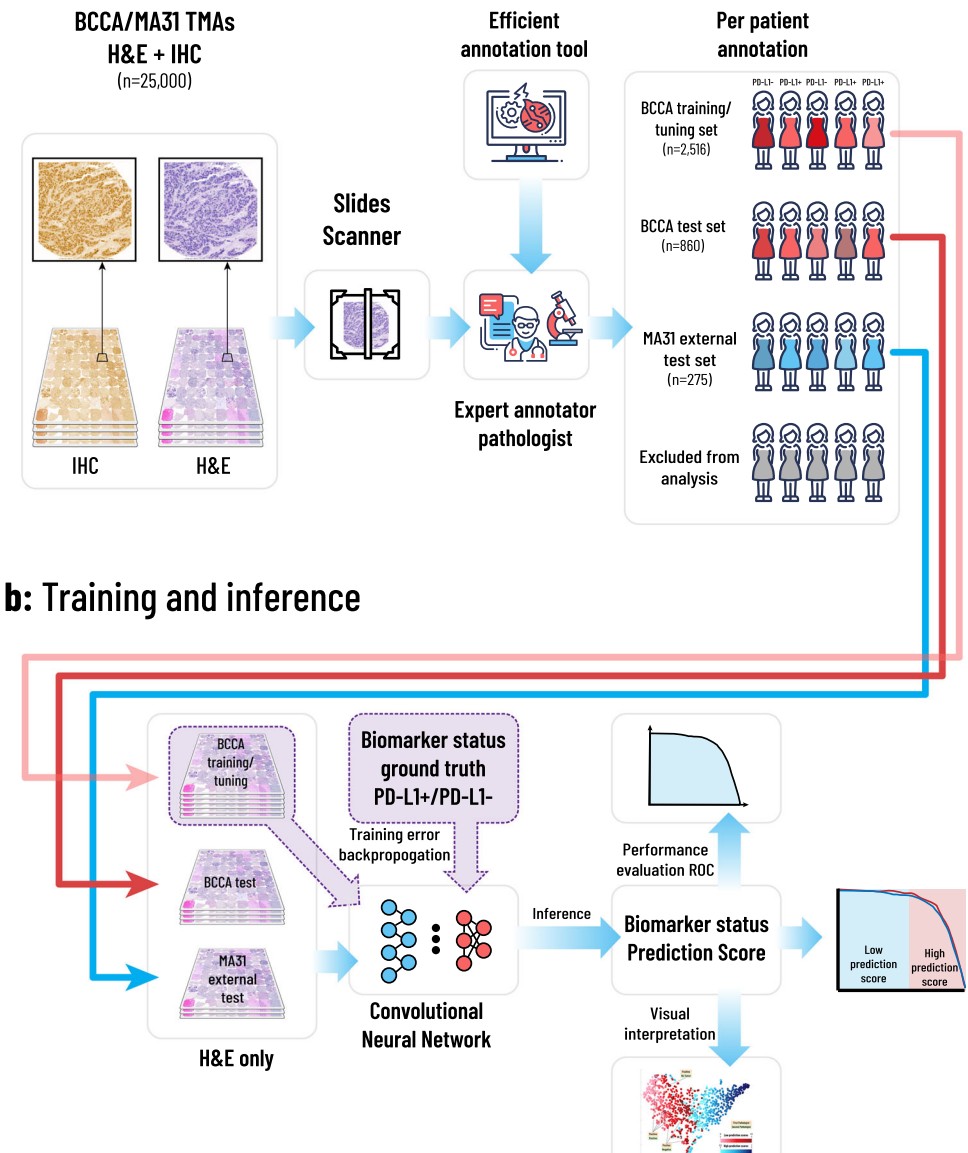

**a: Data annotation process**

**b: Training and inference**

**Fig. 1 | Overview of the proposed framework.** The annotation, training, and inference methodologies. **a** An expert pathologist used our designed computer-aided annotation software to annotate patients for PD-L1 status, based on their H&E and corresponding IHC-stained TMA images. Patients with no TMAs, unclear images, deficient staining, and with insufficient tissue or tumor, were excluded from the analysis. The rest of the patients were assigned each a PD-L1 positive or negative label, resulting in 2516 annotated patients in the BCCA training set, 860 in the BCCA test set, and 275 in the MA31 external test set. **b** H&E images of the included patients were assigned the annotation of their corresponding patients. The H&E images in the BCCA training set were used to train and validate the CNN in a 5-fold cross-validation manner, using the ground truth PD-L1 annotations. The model was then applied to the validation folds, the BCCA test set, and the external MA31 test set, to produce a prediction score for each H&E image. The prediction score per patient was defined as the maximum over its corresponding H&E prediction scores. The prediction scores at the patient level were then compared to the ground truth PD-L1 annotations to produce statistical analyses.

validated using 5-fold cross-validation (CV). In each training phase, the model obtained pre-processed H&E-stained TMA images belonging to patients from the training folds and was optimized to predict the PD-L1 status that was determined by the pathologist. Each H&E image has been assigned the status of its corresponding patient. During the inference phase, the model obtained H&E images belonging to new patients from the validation fold, yet unseen by the system, and produced a prediction score between 0 and 1 for each patient. The prediction scores were then compared to the ground truth annotation of the pathologist for statistical analysis (Fig. 1b). The resulting area under the curve (AUC) performance for the cross-validation with respect to

the pathologist's binary PD-L1 status was 0.911 (95% confidence interval (CI): 0.891–0.925), showing high prediction ability for PD-L1 from H&E images (Fig. 2a, BCCA-CV).

Next, to validate the system on another test set that was not part of the cross-validation, we trained one model with the same architecture, configurations, hyperparameters, and number of epochs, on the entire training set, and applied it to the held-out BCCA test set, resulting in a PD-L1 prediction score for each of the 860 test patients. The resulting AUC performance was 0.915 (95% CI: 0.883–0.937) for the BCCA test set (Fig. 2a, BCCA-test). The test AUC serves as a second validation of the system and shows that the system could be

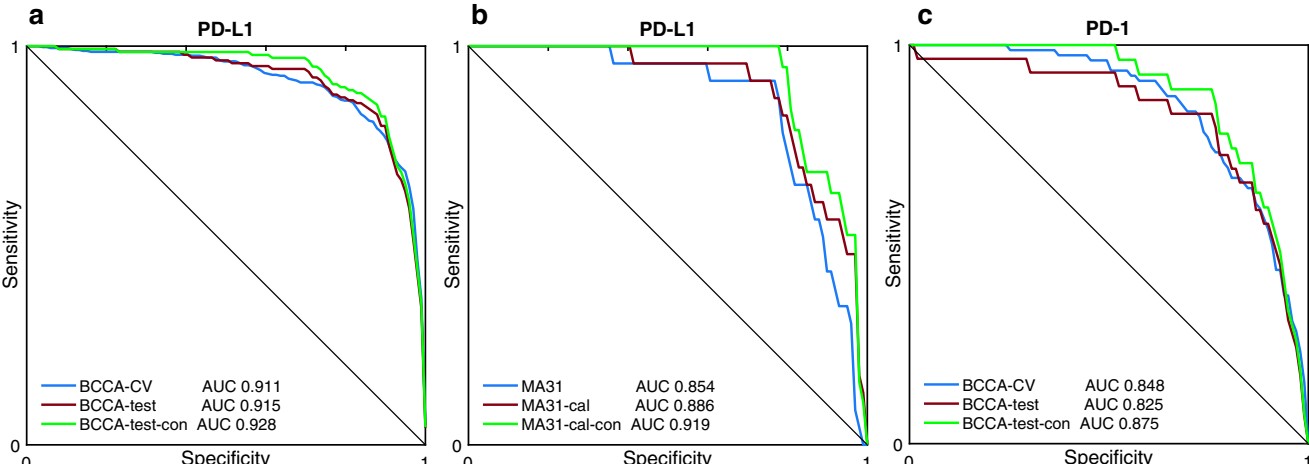

**Fig. 2 | Convolutional neural networks achieve high performance in the prediction of PD-L1 and PD-1 expression.** Receiver operating characteristics (ROC) curves for the performance of the proposed models, in terms of AUC, for PD-L1 and PD-1 prediction in the BCCA and MA31 cohorts. **a** The model obtained high prediction accuracies for both the BCCA cross-validation (0.911) and BCCA test set (0.915). When analyzing only concordant cases between pathologists, AUC performance was further increased (0.928). **b** For the external MA31 cohort, the performance dropped to 0.854, showing that a calibration step may benefit the application of the system to new cohorts. Indeed, the calibration step increased the AUC on MA31 to 0.886, which was further increased to 0.919 after removing the discordant cases. **c** The AUC performance results for PD-1 prediction were lower than for PD-L1. The PD-1 AUC results were high, however, given the extremely imbalanced nature of data (only 3% positives), which poses optimization difficulties due to very few positive samples to train the system with.

generalized to new patients in BCCA that did not take part in the cross-validation. As it is within the confidence interval of the cross-validation AUC, it shows that the model architecture and hyperparameters were not overfitted to the training and validation folds. Given the imbalanced nature of the dataset, we also plot the precision versus recall and negative predictive value (NPV) versus specificity curve for each corresponding ROC curve (Supplementary Fig. 2).

**Prediction on an independent external cohort**

We next aimed to validate the system on an external independent cohort. We applied the same trained model, without any modifications, to obtain a PD-L1 prediction score for each of the 275 patients in the clinical trial MA31 test set. The resulting AUC performance for PD-L1 prediction was 0.854 (95% CI: 0.771–0.908) on MA31 (Fig. 2b, MA31). It is most likely that the low AUC = 0.854 performance on MA31, compared to AUC = 0.915 on the BCCA test set, was due to overfitting of the model to the characteristics of the BCCA cohort, on which it was trained. For example, MA31 had only ERBB2 positive metastatic breast cancer, with 8.4% PD-L1 positive cases, compared to 16.5% positive cases in BCCA. Tissue preparation and staining, and digitization may have also taken part in this overfitting. This drop in performance is a common and expected behavior of deep learning when dealing with a difficult task, even when using extensive data augmentation.

To compensate for these cohort differences and calibrate the model to better fit MA31, we applied a transfer learning approach[28] to fine-tune only the last layer of the CNN using the TMAs from some of the patients (see "Methods"). The patients whose TMAs were used for fine-tuning the model were excluded from the inference and prediction analysis. The calibration resulted in prediction scores that obtained an AUC performance of 0.886 (95% CI: 0.805–0.934) for MA31 (Fig. 2b, MA31-cal), which was higher by 0.036 (95% CI: -0.014 – 0.095) than the AUC of the uncalibrated system. This shows that when applying the model to a new cohort, a calibration step using some samples from the new cohort may increase the performance.

**A decision support system in clinical practice**

Since most patients in the data had low expression for PD-L1, the system could better learn features indicative of negative PD-L1 status. A system with high negative predictive value performance, corresponding to high sensitivity, could be useful for screening out patients

negative for PD-L1, and could allow pathologists to focus their attention on the rest of the cases. Moreover, such a system could be used in clinical practice for quality assurance of PD-L1 expression. In such a setting, biopsy samples that were classified as PD-L1 positive by the pathologists but predicted as very likely to be negative for PD-L1 by the system, could be recommended for a second read.

We set a threshold, such that patients who obtained prediction scores below or above that threshold were classified as low-prediction score (low-PS) or high-prediction score (high-PS), respectively. Low-PS patients were predicted as PD-L1 negative, and high-PS patients were predicted as PD-L1 positive. The threshold was tuned once using the BCCA cross-validation to obtain a large low-PS group while maintaining high sensitivity (Fig. 3a). Using this threshold, The BCCA cross-validation scores were divided to low-PS (57.5%) and high-PS (42.5%) patients. Of the low-PS, only 2.4% of the patients were classified as positive by the pathologist, resulting in a negative predictive value (NPV) of 0.976 and a sensitivity of 0.916 in the BCCA cross-validation (Table 2). This shows that the system could detect patients that were very likely to be PD-L1 negative. In other words, we constructed a system that classifies patients to either low-PS or high-PS based on their H&E images. When the system classifies a new (yet unseen) patient from the BCCA cohort as low-PS, we expect the patient to be PD-L1 negative with an estimated probability of 97.6%.

The system was then applied to patients from the BCCA test set using the same threshold (Table 2). As a result, 473 (55.0%) of patients were classified as low-PS, out of which only 8 (1.7%) were previously classified as positive by the pathologist (NPV = 0.983, sensitivity = 0.943), compatible with our expectations (Fig. 3b). On MA31 (without any calibration), 143 (52.0%) patients were classified by the model as low-PS. Out of these, only 1 (0.7%) was previously classified as positive by the pathologist, resulting in a NPV = 0.993 and sensitivity of 0.957 on the external MA31 test set. This shows that even though the uncalibrated model had lower overall AUC performance on the entire MA31 set, it could still accurately detect PD-L1 negative cases. Using the MA31 calibrated model, 185 (69.0%) of the patients were classified as low-PS, out of which 2 (1.1%) of the patients were previously classified positive by the pathologist, resulting in NPV = 0.989 and a sensitivity of 0.913 on the MA31 cohort. Effectively, calibrating the model to MA31 improved its detection capability of PD-L1 negative patients from 52 to 69%. The above results

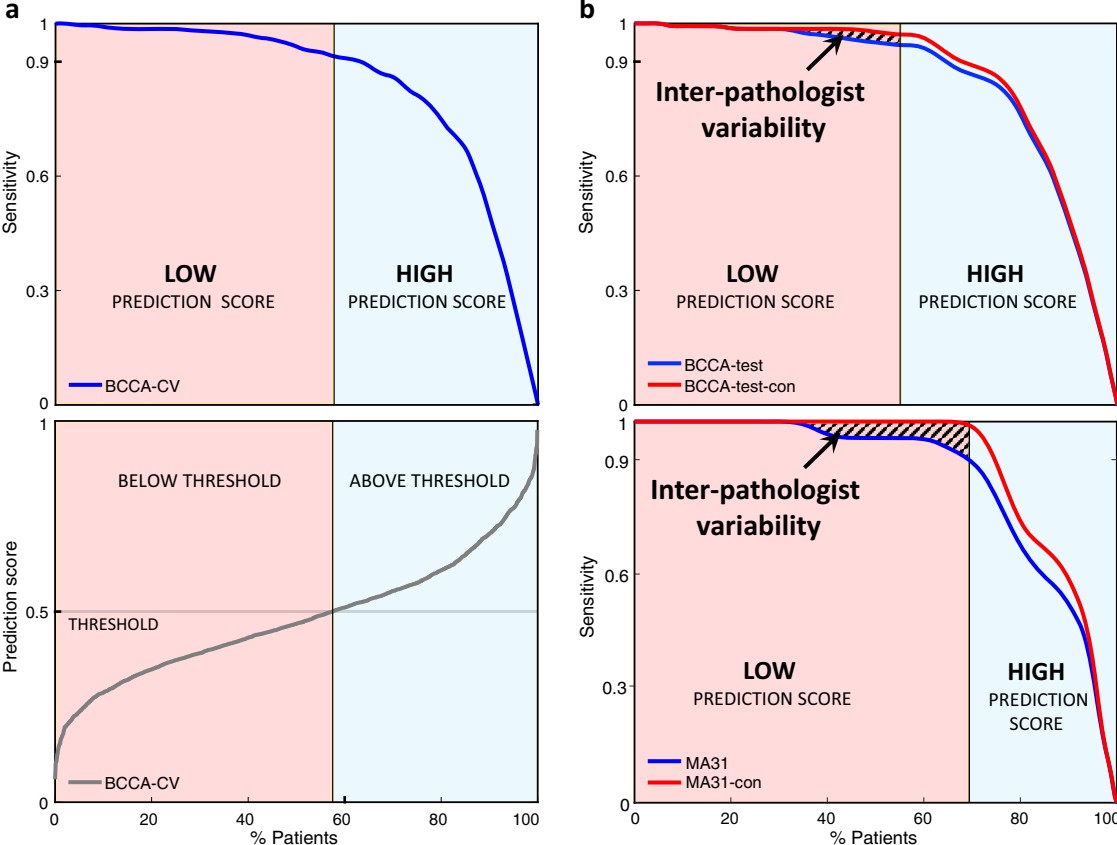

**Fig. 3 | Impact of the proposed system on clinical practice. a** The threshold for splitting the patients' prediction scores to low and high is tuned in the BCCA cross-validation. Bottom: The sorted prediction scores of the patients, versus the percentage of patients classified below the threshold. Top: The cross-validation sensitivity of the system, versus the percentage of patients classified below the threshold (i.e., classified as low-PS), showing a trade-off between the two. The threshold was selected as 0.5, resulting in a sensitivity of 0.92 for BCCA-CV with 58% of the patients in the low-PS group. **b** Applying the selected threshold to the BCCA test patients (top) and MA31 patients (bottom). Following the system's predictions allows the pathologists to focus on reviewing the cases classified as low-PS by the system and positive by the pathologist, which may be prone to miss-interpretation or deficient PD-L1 staining. After removing the discordant cases from the analysis, the sensitivity was increased (BCCA-test-con and MA31-con), revealing the inter-pathologist variability. In addition to quality assurance, the system could be used to allow pathologists to spare IHC staining and interpretation from more than 70% of the patients while retaining 100% sensitivity for PD-L1 expression in MA31.

illustrate that some tumor architectures may be highly indicative of the absence of PD-L1 expression, and that these tumor architectures were reflected in the H&E images and could be recognized by an adequately trained computerized system.

**Interobserver variability and discordant cases**

Previous studies reported low interobserver agreement for PD-L1 quantification by pathologists[10–12]. To gain a better understanding of the system's performance, we next aimed to estimate the inter-observer variability between pathologists for interpretation of PD-L1 expression on our data. Re-annotating the entire BCCA cohort would have been time consuming (estimated 20 h, even with the fast annotation tool), and the MA31 cohort had overall better staining quality than the BCCA cohort. Thus, for this task, we asked a second expert pathologist to re-annotate the entire MA31 cohort, based on the same PD-L1 and corresponding H&E TMA images used by the first pathologist. The second pathologist was blind to the annotations of the first one. Out of the 23 patients that were classified positive in MA31 by the first pathologist, 19 (82.6%) were classified positive by the second one as well, and the remaining 4 were classified negative, discordant from the first pathologist. Out of the 252 patients classified negative by the first pathologist, 246 (97.6%) were classified negative by the second one, and 6 (2.4%) as positive (Table 3a). The Cohen's-kappa concordance between the pathologists was 0.772. Although higher than in the previously mentioned studies (kappa = 0.543–0.628), this shows

that PD-L1 expression can be interpreted differently by independent expert pathologists, and that a supporting system that could add useful prediction information may help pathologists improve their diagnosis and reduce variability.

The proposed system could be applied in clinical practice as an alert system. As such, any case classified positive by the pathologist but low-PS by the system would be alerted, and the alerts would be considered for re-examination. To test this idea, we asked the second pathologist to re-annotate the 8 cases in the BCCA test cohort that were classified positive by the first pathologist and low-PS by the system. To remove bias, we first mixed these cases with another 100 randomly selected cases from the BCCA test cohort. In this manner, the second pathologist was unaware of which of the cases were the ones classified previously as positive or predicted as low-PS. We repeated the process for the MA31 cohort (with the calibrated model). Table 3b summarizes the results, and Fig. 4 shows the reviewed TMAs.

In the BCCA test set, there were 8 low-PS patients that were classified positive by the first pathologist. Only 4 (50.0%) of these cases were also classified by the second pathologist as positive. Among the remaining, 3 cases were classified negative, and 1 case with no tumor to determine. In MA31, there were 2 low-PS patients that were classified positive by the first pathologist. Both cases were classified as negative by the second pathologist. In other words, the two system alerts in MA31 (patients classified positive by the pathologist and low-PS by the system) were indeed both discordant between the pathologists and

**Table 2 | Summary of the system's performance and statistics for PD-L1 and PD-1 in the BCCA and MA31 cohorts**

| Biomarker | Group (+model) | AUC (95% CI) | NPV | Sensitivity | Specificity | Accuracy | # Patients | # Patients predicted as: (%) | | % Positive patients | % Positive patients in: | |
| --- | --- | --- | --- | --- | --- | --- | --- | --- | --- | --- | --- | --- |
| | | | | | | | | Low-PS | High-PS | | Low-PS | High-PS |
| PD-L1 | BCCA-CV | 0.911 (0.891 – 0.925) | 0.976 | 0.916 | 0.672 | 0.907 | 2516 | 1446 (57.5%) | 1070 (42.5%) | 16.60% | 2.40% | 35.70% |
| | BCCA-test | 0.915 (0.883 – 0.937) | 0.983 | 0.943 | 0.646 | 0.900 | 860 | 473 (55.0%) | 387 (45.0%) | 16.30% | 1.70% | 34.10% |
| | BCCA-test-con | 0.928 (0.902 – 0.948) | 0.991 | 0.971 | 0.646 | 0.904 | 856 | 469 (54.8%) | 387 (45.2%) | 15.90% | 0.90% | 34.10%s |
| | MA31 | 0.854 (0.771 – 0.908) | 0.993 | 0.957 | 0.563 | 0.869 | 275 | 143 (52.0%) | 132 (48.0%) | 8.40% | 0.70% | 16.70% |
| | MA31-cal | 0.886 (0.805 – 0.934) | 0.989 | 0.913 | 0.747 | 0.862 | 268 | 185 (69.0%) | 83 (31.0%) | 8.60% | 1.10% | 25.30% |
| | MA31-cal-con | 0.919 (0.864 – 0.952) | 1.000 | 1.000 | 0.757 | 0.876 | 258 | 181 (70.2%) | 77 (29.8%) | 7.40% | 0.00% | 24.70% |
| PD-1 | BCCA-CV | 0.848 (0.807 – 0.883) | 0.998 | 0.974 | 0.422 | 0.966 | 2618 | 1074 (41.0%) | 1074 (41.0%) | 3.00% | 0.20% | 4.90% |
| | BCCA-test | 0.825 (0.697 – 0.890) | 0.994 | 0.913 | 0.377 | 0.966 | 877 | 322 (36.7%) | 555 (63.3%) | 3.30% | 0.60% | 4.90% |
| | BCCA-test-con | 0.875 (0.808 – 0.920) | 1.000 | 1.000 | 0.377 | 0.968 | 875 | 320 (36.6%) | 555 (63.4%) | 3.10% | 0.00% | 4.90% |

MA31-cal stands for the calibrated MA31 model. BCCA-con and MA31-con stand for the analysis after removing cases with PD-L1/PD-1 annotations that were discordant between the pathologists. The specificity states the percent of negative cases that were classified by the system as low-PS (certainly negative), while the sensitivity can be seen as the percent of positive cases that passed the quality assurance. I.e., cases that were not recommended for re-staining or re-interpretation. The trade-off between the sensitivity and specificity is visualized by the corresponding ROC curves (Fig. 2). Note that the specificity can also be interpreted as the probability that the system will detect any random false positive misclassification. This is because it is the probability of a true negative sample, which was erroneously classified as positive, to be in the low-PS group. NPV - Negative predictive value.

were among the four positive cases in MA31 that the second pathologist re-classified as negative. An interesting observation is that if the system alerts were not indicative of potentially discordant cases, the probability for the two alerts to be among the four discordant patients would have been low – 2.37% (calculated as 4/23 for the first alert to be discordant, multiplied by 3/22 for the second alert to be discordant). Also, note that random false positive classifications made by the pathologist would likely be alerted by the system. The probability for a random false positive classification to be alerted can be estimated as the specificity, which is 64.6% on BCCA-test and 74.7% on MA31.

In our data, most of the patients who were classified as low-PS by the system and positive by the first pathologist, were found discordant by the second pathologist. Moreover, it is generally known that IHC staining occasionally suffers from deficiencies. Thus, it is also possible that some of the few cases that were indeed confirmed positive by both pathologists may have, in fact, been PD-L1 negatives that suffered from over-staining or other staining deficiency, and that the system, which based its prediction on the robust H&E staining, could bypass these staining errors. This analysis shows the potential benefit of the system as quality assurance in clinical practice, by detecting cases that could be more sensitive for miss-interpretation and should be reviewed or re-stained again.

The interobserver variability between the two pathologists may also indicate that the ground truth annotation, which the system's prediction was compared to, was not perfect. We removed the data of the patients that the pathologists did not agree on from the analysis (10 cases in MA31, and 4 cases within the low-PS group in BCCA) and re-analyzed the prediction performance. As a result, the AUC performance was improved from 0.915 to 0.928 on the BCCA test set (Fig. 2a, BCCA-test-con), and from 0.886 to 0.919 on MA31 calibrated model (Fig. 2b, MA31-con). Please note that for the BCCA-test data, although an increase in the performance was marked, the new AUC may be biased because cases removed were only from the low-PS positive group. On the BCCA-test set, the NPV was improved from 0.983 to 0.991, and the sensitivity from 0.943 to 0.971 (Fig. 3b, BCCA-test-con). On the MA31 test set, both the NPV and sensitivity improved to 1.0 (perfect score), showing that any patient classified by the system as low-PS and positive by the first pathologist, was interpreted as negative by the second pathologist, in favor of the system (Fig. 3b, MA31). This implies that the system's predictions from H&E images alone may reveal additional information that could guide decisions in clinical practice. Such a system can already be used as a quality assurance and re-annotation recommendation tool for re-interpretation or even re-staining for PD-L1, which does not require extensive validations or FDA approvals.

### PD-1 prediction on the BCCA cohort

The BCCA cohort also contained IHC-stained TMAs for PD-1. We thus repeated the process for the annotation, training, and prediction of PD-1 expression on BCCA. The annotation process of PD-1 on BCCA resulted in extremely imbalanced data, in which almost all (96.9%) of the patients were annotated as negative for PD-1 status (Table 1). We used the same architecture and model for the training and prediction as that of PD-L1. PD-1 prediction was surprisingly high in the BCCA cross-validation (AUC = 0.848) given the low number of positive samples the system could learn from (Fig. 2c). As to patients classified as low-PS (41.0%), only a negligent part (0.2%) was found discordant due to a positive classification by the pathologist (NPV = 0.998, sensitivity = 0.974, Table 2).

In the BCCA test set, the model achieved an AUC close to that of the cross-validation set. Amongst patients classified as low-PS (36.7%), only 2 were designated positive by the first pathologist (NPV = 0.994, sensitivity = 0.931). Both cases were re-classified as negative by the second pathologist, in favor of the system's prediction. Removing these cases increased the AUC from 0.825 to 0.875 and the NPV and

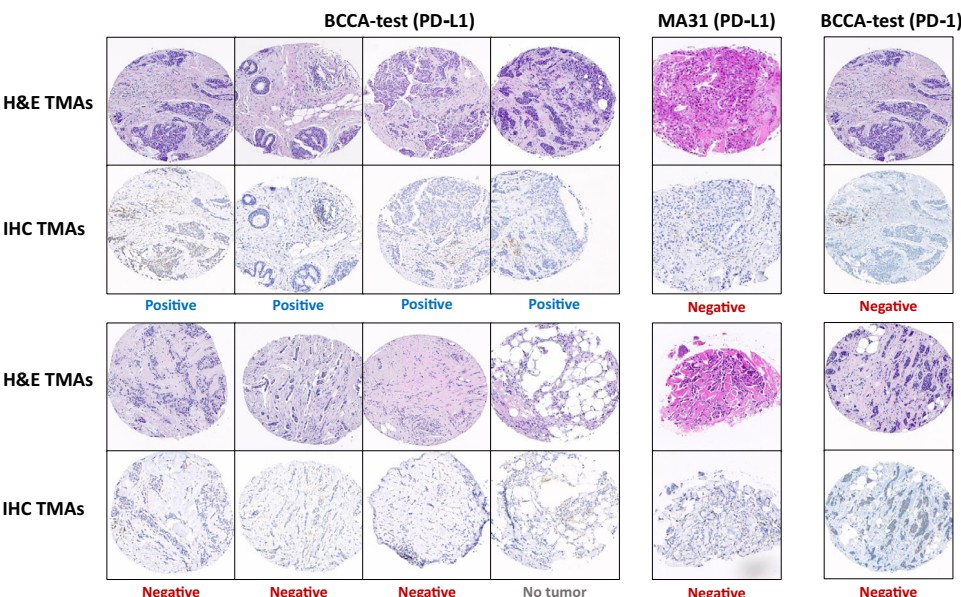

**Fig. 4 | Low-prediction score cases classified positive.** Tissue images of patients classified positive by the first pathologist and low-PS by the system. The BCCA-test patients are shown on the left (by PD-L1) and right (by PD-1), and the MA31 patients are shown in the middle (by PD-L1). For each patient, a representative H&E image and its corresponding IHC image are displayed one below the other. The classification of the second pathologist is registered below each sample, showing that most of the low-PS cases that were classified positive by the first pathologist were classified otherwise by the second one.

sensitivity to 1.0 (perfect score). This shows that although PD-1 had an extremely imbalanced dataset with a very small number of positive samples to train the system on, its prediction performance was relatively high and not much lower than that of PD-L1. Furthermore, the system could detect PD-1 negative patients as accurately as with PD-L1.

## Data interpretation by feature space visualization

t-distributed stochastic neighbor embedding (t-SNE)[29] can be used to visualize the data by mapping patients to points in space, based on their image features. To better understand the correlation between tissue architecture and PD-L1 expression, we applied t-SNE to the feature space representation of the BCCA test TMA images, produced at the inference step of the CNN (See "Methods"). In this case, the t-SNE is optimized to find a mapping such that tissue architectures with similar predictive characteristics for PD-L1 expression are mapped to near points, while dissimilar architectures are mapped to far points. Fig. 5a shows the resulting t-SNE distribution of the patients in 2-dimensional space, colored by their PD-L1 prediction scores. One can see that patients with low and high prediction scores were grouped almost separately, indicating that they have distinctive tissue characteristics. The low-PS cases that were classified as positive by the first pathologist are marked and can be seen to be uniformly distributed within the low-PS group.

Next, we mapped the H&E-stained TMA images of the patients based on the same t-SNE distribution (Fig. 5b), and an expert pathologist compared the TMAs of the different regions in the resulting embedding. TMAs mapped at the bottom of the t-SNE embedding had TMA images with partially missing tissues, showing that their interpretation by the CNN was more distinctive than others, most likely because they did not have enough tissue to obtain a conclusive prediction. Low-PS tumors were characterized by dense streaming desmoplastic stroma surrounding tumor ducts with variously sized lumens, and these ducts were oriented along the stromal fascicles. The tumor-to-stromal ratio was relatively low, and the number of tumor-associated immune cells was low to absent. High-PS tumors were characterized by a crowded population of solid-growing tumor nests, islands with hyperchromatic nuclei, and no lumens. The tissue components, such as stroma and glands, were less structurally oriented

than the low-PS ones. The stroma was haphazardly oriented and hardly streaming, and its area was small with respect to the tumor. The tumor associated-immune cells were present, sometimes in large numbers. Supplementary Fig. 3 shows the corresponding t-SNE with IHC stains, demonstrating that the brown staining, which represents the PD-L1 expression, is concentrated on the high-PS side of the map.

Aggressive tumors, such as triple negative breast cancer, are usually hypercellular, with low stromal to tumor ratio, stroma rich in immune cells, high nuclear to cytoplasm ratio, solid growth pattern, and high mitotic index[30]. It has been shown that PD-L1 expression is correlated with aggressive tumors and with the presence of immune cells[31]. PD-L1 was also shown to be correlated with tumor mutational burden (TMB)[32], which was found to be correlated with high immune cytolytic activity[33]. This could explain the features picked by the algorithm to predict the PD-L1 expression. The visual comparison of the high-PS and low-PS tumors showed that high-PS tumors were indeed hypercellular, had aggressive characteristics, and had more immune cells. The high-PS features were also in line with the finding that TP53 mutation, which is found in aggressive triple-negative tumors[30], can trigger immune response[34] and regulate PDL1 expression[35].

To shed more light on the system's decision making, we analyzed the predictive power of the system for the BCCA-test set within each histologic and tumor subtype, separately (Supplementary Fig. 4). In Supplementary Fig. 4a, we show the t-SNE distribution for each subtype class, by keeping only the embedded points belonging to that class. This analysis showed that the t-SNE distribution was not entirely explained by any of the subtype classes on its own, and that the system was able to accurately predict PD-L1 expression within each of the classes. Both high-PS and low-PS groups contained all possible subtypes. Additionally, the cases that were discordant between the pathologists were uniformly distributed among the subtypes, showing that there was no notable difference in the subtype composition of discordant images compared to non-discordant ones. A comparison of the discordant and non-discordant H&E images of MA31 also showed no notable difference between the features of these two groups (Supplementary Fig. 5). In Supplementary Fig. 4b we show the AUC performance for PD-L1 prediction within each of the classes. The AUC

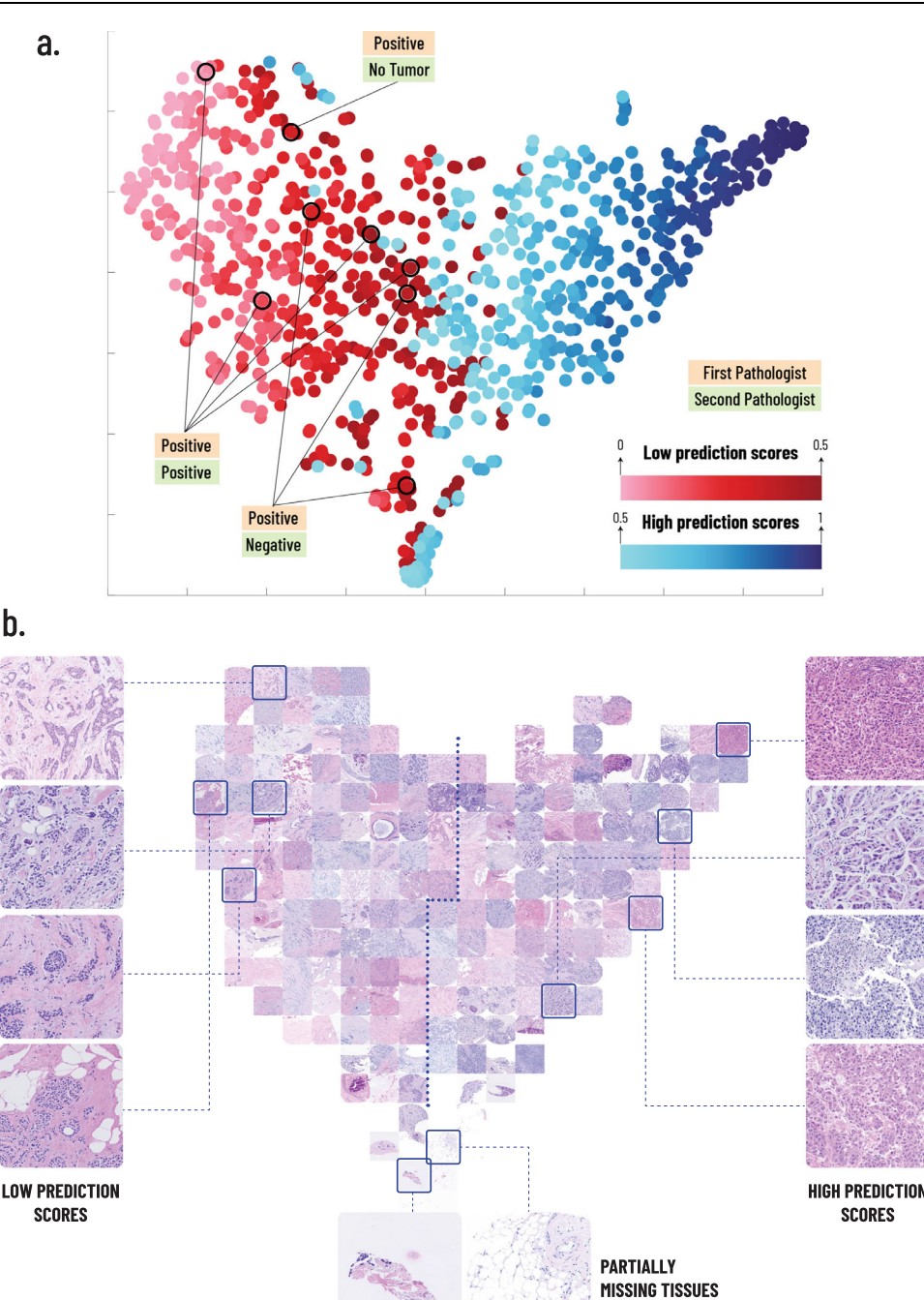

**Fig. 5 | t-SNE embedding for visualization of feature space. a** A 2D visualization of the image feature vectors by applying t-SNE. Each point represents a single patient in the BCCA test set. The t-SNE embedding maps patients with similar image features to near points, and patients with dissimilar image features to far points. The points are colored by the PD-L1 prediction scores of their corresponding patients. The 8 patients that were classified positive by the first pathologist and low-PS by the system are marked and their classifications by both pathologists are noted. **b** The TMA images corresponding to the t-SNE embedding are presented. Several examples of low and high prediction score images are shown, to demonstrate the characteristics observed by the pathologists. Examples of partially missing tissues are shown at the bottom.

of the entire data was within the 95% AUC-CI of any of the subtypes, including the ERBB2 + class, which is in line with the previously shown ability of the system to generalize to the MA31 cohort. For most subtypes, the AUC was lower than the overall AUC. The lowest AUC was obtained for the Luminal ERBB2 + subtype, probably because it consisted of almost only PD-L1 negative cases. The AUC tendency to decrease when constraining to subgroups of patients is an expected outcome, which is due to the existence of correlations between the subtypes and the PD-L1 expression that are explained by the AI-based features.

To complete the analysis, we performed a univariate and a multivariate analysis for PD-L1 prediction based on the AI score and 19 additional molecular biomarkers (See "Methods"). The Univariate analysis shows the correlation between each one of the biomarkers and the PD-L1 status (Supplementary Table 1a). The multivariate analysis demonstrates which biomarkers significantly contribute to the PD-L1 prediction when all biomarkers are used together (Supplementary Table 1b). The AI score had the highest $\chi^2$ value in both analyses. FOXP3 and CD8 tumor infiltrating lymphocytes (TILs) significantly contributed to PD-L1 prediction. This finding is in line with the

correlation between PD-L1 expression and the presence of immune cells and FOXP3 expression[36]. The rest of the biomarkers, although some were significantly correlated with the PD-L1 expression (Supplementary Table 1a), did not further contribute significantly to the combined prediction (Supplementary Table 1b). The overall AUC performance for PD-L1 prediction using both the AI score and all biomarkers was 0.928 (95% CI: 0.897–0.946). The addition of biomarkers to the analysis has significantly increased the AUC ($P = 0.006$). The low-PS group, however, still contained the same discordant cases. This shows that the AI-based features alone could capture almost all molecular information used for PD-L1 prediction. In addition, it can be deduced that a combination of image features with molecular biomarkers has the potential to further increase the ability to assess PD-L1.

## Discussion

Our data and experiments showed that breast cancer tumors have unique architectural signatures that hold information indicative of the expression of PD-L1 and PD-1. These signatures could be revealed by basic H&E staining with an adequate learning system that was trained on pre-annotated examples. Our system showed high prediction abilities for the expression of both PD-L1 and PD-1 based on H&E staining, which is cheaper, more efficient, and more robust than immunohistochemistry staining. Several independent previous studies demonstrated the ability to predict ER, PR, and ERBB2 status from both H&E-stained TMA and WSI images, ER status always obtaining the highest prediction performance[21–25]. One of these studies was done on the same BCCA cohort and predicted Estrogen receptor status with AUC = 0.88. Our study thus revealed that PD-L1 is evidently the biomarker with expression most correlated with tumor architecture in breast cancer, obtaining the highest prediction performance (AUC = 0.91–0.93, Fig. 2a). As an additional verification for this observation, we obtained the ER status data for BCCA, and repeated the training and prediction with the same architecture of our system, this time for ER prediction, and obtained AUC = 0.89.

Our system was trained and validated in a cross-validation manner on the BCCA cohort, while separating patients between train and validation folds. The system was then validated twice more, first on a held-out test set that was not a part of the cross-validation, and then on an external test from the MA31 clinical trial, completely independent from the original BCCA cohort. We showed that the system was able to predict the expression of PD-L1 and PD-1 in all experiments, obtaining slightly inferior results for the independent MA31 test. This outcome could be explained by an unavoidable overfitting of the system to the BCCA cohort on which it was trained, which had different characteristics than MA31 (Table 1). This performance gap can be reduced by significantly enlarging the training data and incorporating several independent cohorts in the training phase. Instead, since our data were limited, we overcame this overfitting by re-calibrating the system to better fit MA31, using transfer learning[28]. The calibration required held-out samples from MA31 that were then excluded from the final test and showed that indeed performance was increased (Fig. 2b).

PD-L1 staining and interpretation are known for having inconsistencies in diagnosis[8–12]. Thus, a quality assurance system that is based on different data, such as H&E images, could improve diagnosis by suggesting second reads for potentially miss-classified cases. To estimate the inter-pathologist variability on our data, a second expert pathologist repeated the annotation process for MA31. Even though MA31 had much better staining quality and was less prone to interpretability errors than BCCA, and even though the agreement between our pathologists was higher than presented in previous studies, PD-L1 quantification by IHC on our data was far from being perfect (Table 3). It is worth noting that CNNs have an inherent ability to be robust to errors in the training data by learning a generalized underlying model. In other words, as long as the errors in the training data are of a reasonable amount, the CNN optimization is almost unaffected by

**Table 3 | a Concordance matrix for the agreement of the two expert pathologists for PD-L1 status in the MA31 cohort, at patient level. b: Re-classification of a second pathologist for PD-L1 and PD-1 status in the BCCA and MA31 cohorts**

**a**

| Patients classified by pathologist #1\#2 as | Negative | Positive | Total | Agreement |
|---|---|---|---|---|
| Negative | 246 | 6 | 252 | 97.60% |
| Positive | 4 | 19 | 23 | 82.60% |
| Total | 250 | 25 | 275 | |
| Agreement | 98.40% | 76% | | |

**b**

| | Classified positive by pathologist #1 | | | |
|---|---|---|---|---|
| | Re-classification by pathologist #2 as | | | Pathologists' agreement |
| Cohort | Negative | Positive | No tumor | |
| **Classified negative by the system** | | | | |
| BCCA-test (PD-L1) | 3 | 4 | 1 | 50% |
| MA31 (PD-L1) | 2 | 0 | 0 | 0% |
| BCCA-test (PD-1) | 2 | 0 | 0 | 0% |

Re-classification was done only for cases classified as negative by the system but positive by the first pathologist.

these errors. Thus, even though the ground truth data may have been imperfect due to the reasons mentioned above, the system has probably learned to predict the true PD-L1 status.

We showed that the system could serve as a decision support and quality assurance in clinical practice. Our system classified patients by their prediction scores to low-PS and high-PS, based on their H&E images. Patients who were predicted as low-PS were very unlikely (0.0–2.4% probability) to be PD-L1 positives (Fig. 3 and Table 2). Low-PS patients who were classified positive by the first pathologist were often re-interpreted as negative by the second pathologist. In fact, the probability of discovering a random false positive misclassification can be estimated as the specificity, i.e., the percentage of low-PS cases out of the negative cases (64.6% on BCCA-test and 74.7% on MA31). This shows that H&E analysis may provide an additional, yet unexploited information, that could guide pathologists' attention to cases that are more prone to miss-interpretation. The quantification of PD-L1 expression depends not only on the pathologists' interpretation but may significantly change due to the choice of antibody and staining method[8,9]. Thus, further study, in which samples can be re-stained, is required to analyze the few remaining cases that were classified as low-PS by the system and positive by both pathologists. Lastly, the results on the independent external MA31 test set showed that roughly 70% of the cases could be ignored while obtaining 100% sensitivity. Thus, a system for PD-L1 status prediction based on H&E could be used to eliminate the need for immunohistochemistry for the majority of the samples, or at least to prioritize those which were among the 30% inconclusive cases.

The lack of interpretability of machine learning poses challenges and complicates supervision of the system[37,38]. Grad-CAM[39] is a commonly used approach for highlighting the image regions the system relied on for making its decision. Nevertheless, the highlighted regions may not always provide a meaningful understanding of the system's decision-making (see supplementary discussion on Grad-CAM). To gain a better understanding of the system's decision process, we applied a t-SNE embedding on the feature space of the H&E image features for visualization of the data in a 2-dimensional space. The t-SNE visualization showed that images classified as low and high PD-L1 expression prediction scores had distinct features. A visual examination of the low-PS and high-PS H&E images by an expert pathologist showed that low-PS tumors were characterized by denser and more oriented desmoplastic stroma than the high-PS tumors. Additionally, tumors classified as low-PS had significantly less tumor associated-immune cells and lower tumor-to-stromal ratio than high-PS tumors. These features were in line with the findings that PD-L1 expression is correlated with aggressive tumors and the presence of immune cells. A subtype and multivariate analysis showed that the AI-based features captured information that was beyond the explainability of any other single molecular biomarker or tumor subtype, and that the additive value of all 19 biomarkers together over the AI score for PD-L1 prediction was only moderate.

PD-L1 expression in breast cancer has gained attention only recently, following its endorsement as a predictive biomarker for immunotherapy response in lung cancer. The effect of PD-L1 expression on breast cancer prognosis has only recently begun to be studied in clinical trials, and large breast cancer datasets, containing H&E and slides and corresponding PD-L1 expression annotations simply do not exist yet. What made this study possible was that we exploited a large tissue microarray repository containing H&E-stained images and multiple corresponding stains for various biomarkers. We invested effort in constructing and organizing two datasets, MA31 and BCCA, out of the repository. With a joint effort of two expert pathologists, aided with a computerized application that we designed for fast annotation, we were able to annotate the entire data. Indeed, one of the drawbacks of this study is that it was conducted on tissue microarray images that have limited clinical translation, rather than whole slide images. And yet, our H&E-stained TMA analysis could accurately predict the PD-L1 expression and even detect instances of discordance between the pathologists. Whole slide images are used in clinical practice and contain much more information than tissue microarrays. They are more challenging to analyze because they require more storage, longer training times, and dealing with artifacts and background segmentations. A review of the previously mentioned studies shows that whole slide image analysis obtains better performance than tissue microarray on different prediction tasks. Thus, one can expect that a future study on prediction of PD-L1 status from whole slide H&E images, upon constructing such a database, can only improve our prediction results. Finally, our system can be applied to whole slide images by automatically selecting tiles from regions of interest[40].

## Methods

### Characteristics of the patients and the stains

The dataset used in this study consists of two independent cohorts: BCCA and MA31 (Table 1). Each cohort contains breast cancer tissue samples and clinicopathological data with TMA images. Each patient in the BCCA cohort had 3 H&E-stained TMA cores, one IHC-stained TMA for PD-L1, and one for PD-1. Each patient in the MA31 cohort had between 1 to 4 H&E-stained images, and one PD-L1-stained image corresponding to each H&E image. An expert pathologist annotated the data for PD-L1 positive or negative status, by going through all available H&E and IHC-stained TMA images (Fig. 1a). Some of the samples were annotated to be excluded from the analysis (Table 1), while the rest of the patients were classified as either negative or positive for PD-L1 status. BCCA median follow-up was 12.4 years, and age at diagnosis 62 years. MA31 median follow-up was 21.5 months, and mean age at diagnosis was 55 years. The TMA images from both cohorts contain 0.6-mm-diameter cores and were scanned using the Bacus Laboratories, Inc. Slide Scanner (Bliss) scanner at a resolution of $2256 \times 1440$ pixels.

### Computer aided application for fast annotation

To enable a fast and accurate annotation procedure, we created a computer-aided application for PD-L1 and PD-1 annotation, based on an interactive visualization of each patient's H&E and IHC-stained TMA images. A dedicated button was used to swap between the patients, and between H&E and corresponding IHC images of the same patient. For each case, the pathologist could press one of seven dedicated buttons to choose between "Negative", "Positive", "No TMA", "No tissue", "No tumor", "Deficient staining" and "Out of focus". When pressing a label button, the display window was immediately colored with a distinct color to visualize the selected label and thus prevent mislabeling errors. Additional buttons were designated for navigating to previous patients, inserting comments for specific cases, and for saving the inputs, to continue the annotation process later. To avoid undesired bias, the pathologists did not have any access to metadata such as the patient IDs.

### Assessment of PD-L1 and PD-1 expression

PD-L1 expression was determined using the Ventana PD-L1 (SP142) assay as the proportion of tumor area occupied by PD-L1 expressing tumor-infiltrating immune cells (IC), and an expression in ≥1% IC was defined as PD-L1 positive status, in accordance with the FDA guidelines[7,41]. PD-1 expression (Ventana NAT105) was determined using the combined positive score (CPS), corresponding to the proportion of PD-1-stained cells (tumor cells, lymphocytes, macrophages) out of the total number of viable tumor cells, and was considered positive if CPS ≥ 1%.

### Data pre-processing and augmentation

First, a square section of $1440 \times 1440$ was cropped from the center of each TMA image. During training, the squared section was randomly

cropped from within the 1640 × 1440 middle section of the image, while during inference, it was fixed to the center 1440 × 1440 pixels of the image. This random crop mechanism was designed to make the model more robust to the variance in the location of the tissue in the scanned slides, which is not always perfectly centered. Then, all images were resized to a resolution of 512 × 512. Next, we performed data augmentation to help the model deal with variability in staining methods and between the cohorts. Specifically, we randomly flipped and rotated each image, and performed color jittering. The color jittering included gamma correction, jittering the saturation and temperature of the image, additive Gaussian noise, and blur augmentations. No data augmentations were performed during inference.

## Convolutional neural network architecture

Inspired by the ResNet[42] architecture, we used a CNN with residual connections to train and test the model (Supplementary Fig. 1). The CNN model consists of an initial convolution layer followed by a ReLU activation and a BatchNorm layer. Then, a CNN backbone that consists of 4 blocks encodes the input to an embedding vector in the latent space of $R^{256}$. Each block in the CNN contains a spatial downsample operation, implemented by a convolution layer with stride 2, followed by three convolution layers with stride 1. Each convolution layer was chosen with a kernel of size 3 × 3, followed by a ReLU activation function and a BatchNorm layer. Following the downsample operation, a residual skip connection adds the input and output of each block. At the end of the last block, we added an Average Pooling layer to reduce the spatial dimensions to a single embedding vector of size 1 × 256, which is the final descriptor of the input image. The embedding vector was then classified to a specific label using a Linear layer of size 256 × 2. The output vector of size 2 contained the final scores for each label and was then passed to a SoftMax activation to compute the final class probabilities. The complete model contains 2,755,538 trainable parameters.

Due to the imbalance in the data labels (see Table 1), we used Focal Loss[43] as the training target function. An undesired bias may occur when unbalanced train data are presented to a deep learning training pipeline using the commonly used cross-entropy loss. The network will be biased toward learning the dominating negative class rather than the actual distribution of the data. Focal Loss tackles this problem by reshaping the standard cross-entropy to scale down the loss assigned to well-classified examples. Thus, the Focal Loss focuses the training on hard examples and prevents the dominating number of easy negatives from overwhelming the network during training, achieving better generalization, and limiting undesired bias.

## Training configurations

We used the same hyperparameters in all experiments. We trained each model using RAdam[44] optimizer, with a batch size of 32, for a total of 110 epochs. We used a learning rate of 0.001 in the first 80 epochs, followed by 30 epochs with a learning rate of 0.0001. For the focal loss, the gamma value was set to 3. Any data split to either validation, training, test, or calibration sets, was always performed per patient. For example, if a TMA image of some patient was included in the test set, any other information of that patient was also included in the test set, and never seen by the system during training. To obtain a prediction score per patient, we considered the maximum of the scores of its H&E images. After the cross-validation stage, the final model was run once on the BCCA test set and the MA31 set to estimate generalization performance. To calibrate the system to perform better on the MA31 dataset, the MA31 set was randomly split at the patient level into calibration (50%) and test (50%) sets. We used the calibration set of patients to fine-tune the last layer of the CNN, following the transfer learning approach[28], and then applied the calibrated model to the test patients. The calibration experiment was repeated 5 times and scores were averaged per patient. The threshold for classifying patients to low-PS and high-PS was determined once during the cross-validation and was set to 0.5 for PD-L1 and to 0.28 for PD-1.

## Statistical analysis

Data were collected, annotated, and analyzed from July 1, 2015, through February 1, 2022. We used the area under the curve (AUC), Cohen's kappa[45], specificity, sensitivity (=recall), positive predictive value (=precision) and negative predictive value (NPV) as our statistical measures. The receiver operating characteristics (ROC) curves were plotted as sensitivity vs specificity. The confidence intervals were computed using bootstrapping. $P < 0.05$ with a 1-tailed hypothesis test indicated statistical significance. The accuracy was measured after binarizing the scores with a threshold that was optimized using the BCCA training data. The Univariate analysis was done by fitting a logistic regression for each variable separately for prediction of the PD-L1 status. The multivariate analysis was done by fitting an $L1$-regularized logistic regression for all variables together using the BCCA training data. The regularization was chosen as the one obtaining the optimal fit, and then the model was applied to the BCCA test set. The statistical analysis was implemented using Matlab (R2019a).

## Visualization by t-SNE distribution

To create the t-SNE visualization, for each H&E-stained TMA image we extracted the features of the last layer of the CNN during the inference step. Then, each patient was represented by the feature vector of its H&E image that obtained the highest prediction score. t-SNE was then applied to the feature vectors to obtain a 2D representation for each patient. The deconvolution into hematoxylin and DAB channels in Supplementary Fig. 3 was done using the unmixing model[46].

## Hardware and software

All data were stored and processed on our in-house servers, graphics processing units (GPUs), and central processing units (CPUs): a cluster of 14 Intel Xeon CPUs and 30 1080ti and 2080ti GPUs scheduled by a Slurm system with NAS storage. We used Python 3.7 with Pytorch 1.8 to train and test the models. Each model was trained on a single GPU. All code was developed using open-source tools and packages.

## Ethical review

All research at the Genetic Pathology Evaluation Centre is performed in accordance with institutional and provincial ethical guidelines. Because the data did not include patient contact or medical record review, informed consent was not required.

## Reporting summary

Further information on research design is available in the Nature Research Reporting Summary linked to this article.

# Data availability

The database was composed from a publicly available tissue micro-array (TMA) library, published by the Genetic Pathology Evaluation Centre. The TMA datasets can be downloaded from http://bliss.gpec. ubc.ca by navigating to 02-008 for the BCCA cohort and to MA31 for the MA31 cohort. Source data are provided with this paper.

# Code availability

Our code and experiments can be reproduced by utilizing the details provided in the "Methods" section on data pre-processing and augmentation, model architecture, and training configurations. Data pre-processing is based on imgaug library (https://github.com/aleju/imgaug). The components of our model architecture and training protocol can be reproduced using ResNet (https://pytorch.org/vision/0.8/_modules/torchvision/models/resnet.html) and Focal Loss (https://pytorch.org/vision/stable/_modules/torchvision/ops/focal_loss.html).

Our Computer aided application for fast annotation is available at https://github.com/amirlivne/PD-L1_Annotator. Our trained model is also available in https://github.com/amirlivne/PD-L1_predictor.

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

## Acknowledgements

This research was supported by the Israel Innovation Authority – Kamin 69997 (R.K), the Israel Science Foundation (ISF) grant 679/18 (R.K), and the Israel Precision Medicine Partnership program (IPMP) grant 3864/21 (R.K). We would like to thank Alon Zvirin for proofreading the paper, and Shakeel Virk and Samuel Leung for helping with the data acquisition. None of these contributors were compensated.

## Author contributions

G.Shamai. designed the concept and experiments. G.Shamai. and A.L. constructed the machine learning dataset. A.P. and E.S. performed the data annotations. A.L. performed the machine learning experiments and designed and implemented the annotation software. G.Shamai. and A.L. performed the statistical analysis. E.S. and A.C. performed the heatmap interpretation. G.Shamai. and R.K. supervised and managed the project. All authors contributed for drafting the paper and for critical review.

## Competing interests

The authors declare no competing interests.
