## [Peer Review File · Nature Communications]

Reviewers' Comments:

Reviewer #1:

Remarks to the Author:

This paper presents a classical CNN based classification model to classify breast cancer image patches into PD-L1/PD1 positive and negative groups. Although the idea and logic are clear and easy to follow, the overall study is very simple and lacks potentially insightful explorations. The delivered knowledge is unclear and not supported by the results. Several questions remain to be clarified and investigated.

1) This method was focusing on TMA. Does this method work on WSI? Can the study be applied to another surgical resection dataset, such as TCGA (which is a public dataset), to evaluate the predicted PD-L1/PD1 expression vs. RNA expression?

2) What's the percentage of non-tumor or empty patches in training/validation/testing set? Were they removed for an unbiased evaluation in the validation/testing set?

3) In Figure 4, there is obviously positive brown staining in the last 2 columns for the 1st row and 1st & 6th columns for the 2nd row. It seems that the evaluation from the 2nd pathologist is questionable.

4) In Figure 5b, please put corresponding IHC images together with HE images for the highlighted examples and/or make another tSNE plot with the corresponding IHC images on the same position. It is even better if the IHC images can be deconvoluted into DAB channels therefore the visualization will be easier.

5) Although the main text stated that "low-PS tumors have denser and more oriented stroma; low-PS tumors have significantly less immune cells and higher tumor to stromal ratio", such significance is not supported by the results. The knowledge from the provided examples in Figure 5b even contradicts with such statement: the low-PS tumors from the 4 examples obviously have lower tumor to stromal ratio than the 4 high-PS tumor examples.

Moreover, in Figure 5b, the HE images look more blue in the predicted high-PS score group. it is unclear which information the model is utilizing; i.e., it is unclear whether the model looks at staining difference or tumor percentage or something else. Therefore, please utilize a deep learning interpretation method, such as GradCAM, to visualize which area the model is focusing on.

6) How can pathologists/researchers utilize the model? Will the scripts and model weights be publicly available?

7) Please add specificity and accuracy to Table 2 as well as the main text.

Reviewer #2:

Remarks to the Author:

Key results: outstanding features of the work

* The authors propose a deep-learning approach to estimate the PD-L1 and PD-1 status of breast cancer tissues from TMA, reaching AUCs between 0.82 to 0.93.

Originality and significance:

The main strength and interesting points in the paper are:

- the performance achieved by the approach
- the size of the datasets is important and an additional external cohort is available, strengthening the claims
- the visual analysis of the TSNE to try to understand what features were used by the classifier is quite interesting (though I wish it would be discussed a bit more).

The main weaknesses of the paper are:

- The code does not seem to be made public, and the details of the architecture are not enough to allow anyone to re-code and re-produce it potentially
- Lack of references to the field

Validity:

* The methodology seems to be in line with recent publications done in the field of histopathology AI. However, the code has not been made available so no comment on reproducibility can be made. Any reason the authors do not make it available to the community?

Data & methodology:

- * Line 20: " was optimized to predict the PD-L1 status" - could you tell more about the optimization and which parameters were optimized?
- * Line 156: " significantly increase" - Is it statistically significant?
- * Line 216: For the 100 randomly selected cases, how was the concordance between the 2 experts? It seems like correcting the analysis only for the cases where the AI's decision is different from the expert #1 is biasing the correction.

Conclusions:

- * " Finally, our system can be applied to whole 380 slide images by automatically selecting tiles from regions of interest" - Why not try on some images available from TCGA database?
- * I like the " Data interpretation by feature space visualization" section. Any comment on whether those features make sense, or have any biological meaning re what is known of those mutations and type of cancer?

References: The manuscript seems to lack references and would benefit from being put more in perspective to other existing work.

For example, although done on a different cancer type, I don't see discussion re Puladi, Behrus, et al. "Automated PD-L1 Scoring Using Artificial Intelligence in Head and Neck Squamous Cell Carcinoma." *Cancers* 13.17 (2021): 4409

How does your approach differs from that one and why a different one would be needed for breast cancer?

Other papers in the field seem to be ignored - A few examples:

Kapil, Ansh, et al. "Deep semi supervised generative learning for automated tumor proportion scoring on NSCLC tissue needle biopsies." *Scientific reports* 8.1 (2018): 1-10.

Widmaier, Moritz, et al. "Comparison of continuous measures across diagnostic PD-L1 assays in non-small cell lung cancer using automated image analysis." *Modern Pathology* 33.3 (2020): 380-390.

...

Also, authors may want to look at that recently published paper:

Jin, Weiqiu, and Qingquan Luo. "When artificial intelligence meets PD-1/PD-L1 inhibitors: Population screening, response prediction and efficacy evaluation." *Computers in Biology and Medicine* (2022): 105499

Reviewer #3:

Remarks to the Author:

This paper introduces a new artificial intelligence deep-learning framework for identifying PD-L1 status from hematoxylin and eosin-stained (H&E) tumor microarray images. The proposed model was validated on an external dataset to demonstrate generalizability. While the authors demonstrate for the first time the ability to identify PD-L1 status from H&E stained images, the overall technique is relatively straightforward given the previously published deep learning frameworks used to identify estrogen receptor, progesterone receptor and human epidermal growth factor receptor 2, etc. Understanding this fact, the authors seek to demonstrate the translational relevance of their work as a tool that serves to reduce the challenges of PD-L1 assessment and act as a clinical decision support tool for pathologists.

As deep learning models for the identification of biomarkers in H&E stained images continue to proliferate, the originality of this paper is derived from its proposed ability to serve as an immediately applicable decision support tool by showing its performance on cross validation and an external dataset. The authors focus on the generalizability of their model and argue that the model's ability to gain information from features within the TMA images allow it to have criteria for making its predictions that are less error prone than human assessment (ground truth). However, the authors base this conclusion on an exceedingly small proportion of their data—two cases in which the model disagreed with the first pathologist and agreed with the second pathologist. The

significance of the model's agreement with only one of the pathologists is questionable considering that there were four cases in which both pathologists had the same agreement and the model disagreed. More robust computational explanations would be welcome.

Specific Comments:

1. (95) "This cohort consists of 652 recruited patients with ERBB2-positive metastatic breast cancer"

The authors assume that model validation on this cohort of patients remains valid despite the population consisting of only patients with ERBB2-positive, whereas the training data consists of all breast cancers. While this is perhaps a valid assumption, authors should address this in their analysis or discussion. For example, is there an association between ERBB2 positive and PDL1 positive in the validation set? Or between triple positive and PDL1 positive in the training set?

2. (190) "NPV = 0.989 and a specificity of 0.913"

This is somewhat confusing, as specificity is not reported in Table 2. Authors should also compare specificity.

3. (242) "On the MA31 test set, both the NPV and sensitivity improved to 1.0 (perfect score), showing that any patient classified by the system as low-PS and positive by the first pathologist, was interpreted as negative by the second pathologist, in favor of the system (Figure 3b, MA31). This implies that the system's predictions from H&E images alone may reveal additional information that could guide decisions in clinical practice."

The authors might soften this claim given the low number of applicable patients.

4. (277) "One can see that patients with low and high prediction scores were grouped almost separately, indicating that they have distinctive tissue characteristics."

This analysis is beneficial and adds to the explainability of the model. However, are there other factors (triple-positive, date of diagnosis, etc) which explain this distribution? Additional efforts to demonstrate the model's explainability could serve to provide additional justification for the previous claims that the model can have superior robustness to cases where there is pathologist disagreement. What are the features that lead to disagreement between the model and the pathologist, or between pathologists? Can these features themselves be used to identify images that should undergo additional pathological review or be restained?

5. (322) "The calibration required held-out samples from MA31 that were then excluded from the final test and showed that indeed performance was increased (Figure 2b)."

This was somewhat confusing? What is the value of calibration when the testing set is then reduced in size—won't reducing the size of the test set and essentially increasing the number of training images always lead to increased fit?

6. Given the imbalance in the data, Precision-recall should be reported in parallel with ROC throughout.

To all reviewers

Please find below a point-by-point response to all of your comments referring to the relevant changes made in the manuscript. We also attach a copy of the revised manuscript with all modifications in color for your convenience.

Reviewer #1 (Remarks to the Authors, Answers)

This paper presents a classical CNN based classification model to classify breast cancer image patches into PD-L1/PD1 positive and negative groups. Although the idea and logic are clear and easy to follow, the overall study is very simple and lacks potentially insightful explorations. The delivered knowledge is unclear and not supported by the results. Several questions remain to be clarified and investigated.

1) This method was focusing on TMA. Does this method work on WSI? Can the study be applied to another surgical resection dataset, such as TCGA (which is a public dataset), to evaluate the predicted PD-L1/PD1 expression vs. RNA expression?

Unlike TMAs, WSIs are more challenging to handle. Each slide image is of size 1-3GB, and contains not only a tumor but also stroma, background, and artifacts one has to deal with. The corresponding scans require more storage than TMAs, stronger GPUs, and much more time to process. Since the slides are too large for CNNs to process as a whole, and since their size varies, one has to split them into small patches. And yet, various recent studies have shown that prediction tasks can be extended from TMAs to WSIs, where the above challenges are dealt with. Thus, our method can be extended to WSIs using an existing framework for processing WSIs. Moreover, as previous studies have shown, the prediction ability increases when using a WSI instead of small TMA samples of the same biopsy.

The problem, in our case, is that there are no such datasets, at least that we know of, that contain a large cohort of WSI with PD-L1/PD-1 expression by IHC for breast cancer. This is why our data, although consisting of TMAs, is the first to show that PD-L1/PD-1 prediction is possible. The TCGA data, which is publicly available, contains H&E-stained breast cancer WSIs with corresponding PDL-1 expression, but only by RNA expression. Although we expect AI to predict the PD-L1 RNA expression, our system cannot be generalized to this prediction on new data because a calibration step would be needed (as we did on MA31), and the calibration would need the PD-L1 by IHC. On TCGA, this calibration is even more crucial than on MA31, because of the larger difference between the datasets (from TMA cores to WSI patches). In addition to the missing data for the calibration step, the PD-L1 RNA expression is only moderately correlated to PDL1 expression by IHC, and thus we do not expect a system that predicts PD-L1 by IHC to predict PD-L1 by RNA. Finally, PD-L1 quantification is only approved by the current guidelines using IHC and not RNA expression, and it is not even clear which cut-off point should be used for binarizing PD-L1 by RNA expression. For these reasons, we chose not to analyze the TCGA data in our paper.

Although the TCGA WSIs analysis by RNA expression may not be relevant, we did this (quite large) experiment at the request of the reviewer and to show that the extension to WSI is practical. We downloaded the BRCA TCGA data and the corresponding RNA expression for PD-L1. We segmented the background of the slides and divided each slide to patches with the same size and magnification as our TMAs. We applied our trained system to the patches with the calibration step using the RNA expression. To binarize the PD-L1

expression we used the expression quantile that would lead to the same percentage of PD-L1 positives as in the BCCA training data. As a result, the AUC performance for PD-L1 expression by RNA was 0.607 (95% CI: 0.555 – 0.659) without calibration and 0.660 (95% CI: 0.531 – 0.772) with calibration. This was an encouraging result given that the concordance between PD-L1 by IHC and RNA is only 54.6-71.2% (PMID: 34572882, breast cancer), and given that the TCGA WSIs' quality is only moderate.

2) What's the percentage of non-tumor or empty patches in training/validation/testing set? Were they removed for an unbiased evaluation in the validation/testing set?

The first pathologist annotated the BCCA and MA31 patients and gave his inputs regarding cases we should exclude. Following this annotation, cases with no tumor/no tissue were excluded from the analysis, because they did not have ground truth labels and their analysis would not be valid. In the BCCA cohort, 969 out of the 4,944 (19.6%) patients had no tissue, and 176 (3.6%) patients had no tumor. In the MA31 cohort, 118 out of the 652 (18.1%) patients had no tissue, and 25 (3.8%) patients had no tumor (these numbers can be deduced from Table 1). Here are some examples:

Only after excluding the cases, we split the remaining included patients into validation, train and test sets. Thus, the proportion of non-tumor/non-tissue patches was independent of the data split, and they were not removed for bias purposes.

3) In Figure 4, there is obviously positive brown staining in the last 2 columns for the 1st row and 1st & 6th columns for the 2nd row. It seems that the evaluation from the 2nd pathologist is questionable.

The pathologist was aware of the brown color in the mentioned cases while performing the evaluation. The presence of brown color is not enough for determination of positive expression. According to the Ventana PD-L1 SP142 assay interpretation guide for breast carcinoma, a true positive staining of the immune cells is considered when the staining is dark, granular, or beaded, rather than smeared, which may indicate a non-specific artifact. As for tumor cells, PD-L1/PD-1 positive staining should be considered only when a membrane staining is seen rather than a non-specific weak cytoplasmic staining. Moreover, according to the clinical guidelines, the threshold for PDL1 positive score is 1% immune cells relative to the number of tumor cells. Thus, a brown spot is not sufficient to make the score positive, and an expert pathologist is required to determine a true positive staining. We would also like to note that both pathologists are experts with life time experience in the breast cancer field. In addition, their evaluation had a high overall concordance (higher than average, according to other reported numbers in all papers we cited), which also shows the validity of their annotations. Lastly, please note that a misclassification of an expert pathologist is exactly what justifies the use of a quality assurance system.

4) In Figure 5b, please put corresponding IHC images together with HE images for the highlighted examples and/or make another tSNE plot with the corresponding IHC images on the same position. It is even better if the IHC images can be deconvoluted into DAB channels therefore the visualization will be easier.

Thank you for your suggestions. We created a corresponding TSNE plot for the IHC images. We also deconvoluted the highlighted example images, as suggested. To keep the main figures concise and for better visualization, we kept Figure 5 as is, and added the new TSNE as Supplementary Figure 3.

5) Although the main text stated that “low-PS tumors have denser and more oriented stroma; low-PS tumors have significantly less immune cells and higher tumor to stromal ratio”, such significance is not supported by the results. The knowledge from the provided examples in Figure 5b even contradicts with such statement: the low-PS tumors from the 4 examples obviously have lower tumor to stromal ratio than the 4 high-PS tumor examples.

Thank you for noticing. For low-PS there was less tumor, and more stroma, so the ratio of tumor to stroma is **lower** and not higher. This was indeed a typo in the discussion text, and it should be written “lower” instead of “higher”. We fix the text in line 413. Please note that in the main text of the paper (results section, line 298), the text was correct.

Moreover, in Figure 5b, the HE images look more blue in the predicted high-PS score group. it is unclear which information the model is utilizing; i.e., it is unclear whether the model looks at staining difference or tumor percentage or something else. Therefore, please utilize a deep learning interpretation method, such as GradCAM, to visualize which area the model is focusing on.

The high-PS images had more tumor and were more hypercellular than the low-PS ones. Since nuclei are stained in blue, the high-PS images looked more blue than the low-PS ones in the t-SNE embedding. Nevertheless, this was not always true because some low-PS images were more blue than some high-PS images, showing that the color itself could not be the reason for decision:

In essence, the annotation by our pathologists was done only after the staining and digitization process, and thus colors that stem solely from the type of scanner or staining could not be correlated to the PD-L1 status. In addition, the performance of the system was high on validation and two test sets and not only on the same data it was trained on, which proves the prediction ability of the system to generalize to new data, meaning that if the tissue color is one of the decision factors, it should be a legitimate factor that can be relied on. Note that we used color augmentation in our training data to increase the generalization

by mitigating sensitivity of the network to non-correlated color variations. Finally, it is worth noting that even if the system could perfectly predict the PD-L1 status, this does not mean that the features the system relies on could always be interpretable.

Regarding the GradCAM visualization, although it can highlight the areas on which the system relies on, we did not find them very valuable for better understanding the features or the decision of the system, and we chose not to present the GradCAM maps in the paper. Below we present a few examples of GradCAM maps of PD-L1 of low-PS and high-PS stains in the BCCA-test cohort. The blue/red colors indicate regions correlated to negative\positive PD-L1 status, respectively.

The GradCAM maps may be too coarse for the size of the cells and TMAs in our context. Also, we expect the highlighted areas to be around the tumor cells expressing PD-L1 for positive examples, and more vague and non-informative for PD-L1 negative samples with no such cells. Since the system mostly focused on prediction of PD-L1 negativity and not positivity, this could also explain why the highlighted regions were less informative. Instead, to better understand the features the system relied on, we performed a few other experiments. Please see the new text in the “Data interpretation by feature space visualization” section.

6) How can pathologists/researchers utilize the model? Will the scripts and model weights be publicly available?

Thank you for the comment, we added a section for the code availability in which we added links for all the components necessary for constructing the model. We included a github link to our computer aided application for fast annotation. We also included a link for downloading our trained model that will be publicly available (“Code Availability” section).

7) Please add specificity and accuracy to Table 2 as well as the main text.

Thank you for your suggestion. This indeed has an interesting interpretation. We added specificity and accuracy to Table 2 and a short interpretation in the caption.

Reviewer 2 (Remarks to the Authors, Answers)

Key results: outstanding features of the work

* The authors propose a deep-learning approach to estimate the PD-L1 and PD-1 status of breast cancer tissues from TMA, reaching AUCs between 0.82 to 0.93.

Originality and significance:

The main strength and interesting points in the paper are:

- the performance achieved by the approach
- the size of the datasets is important and an additional external cohort is available, strengthening the claims
- the visual analysis of the TSNE to try to understand what features were used by the classifier is quite interesting (though I wish it would be discussed a bit more).

The main weaknesses of the paper are:

- The code does not seem to be made public, and the details of the architecture are not enough to allow anyone to re-code and re-produce it potentially
- Lack of references to the field

Validity:

* The methodology seems to be in line with recent publications done in the field of histopathology AI. However, the code has not been made available so no comment on reproducibility can be made. Any reason the authors do not make it available to the community?

Thank you for the comment, we added a section for the code availability in which we added links for all the components necessary for constructing the model. We included a github link to our computer aided application for fast annotation. We also included a link for downloading our trained model that will be available publicly ("Code Availability" section).

Data & methodology:

* Line 20: " was optimized to predict the PD-L1 status" - could you tell more about the optimization and which parameters were optimized?

In deep convolutional neural networks, the image features extracted by the model are computed in several layers of non-linear and linear operations. A set of parameters defines each operation, and the network architecture defines the number of layers, number of parameters, loss, and type of operations. A back propagation process is then applied to optimize these network parameters such that the final feature computation would result in optimal features, in terms of the training loss. In other words, in deep learning the extraction of image features is learned rather than arbitrarily defined (as in shallow learning). In the "Methods" section, we described our model architecture which defines the parameters that are optimized.

* Line 156: " significantly increase" - Is it statistically significant?

Thank you for noticing the misleading terminology. We tested the probability of the null hypothesis and indeed, the increase in the AUC was not "statistically" significant (P value = 0.09). The calibration has resulted in an increase of the AUC by 0.036 (95% CI: -0.014 –

0.095). We indicated the CI of the additive AUC and removed the word “significantly” (line 162).

* Line 216: For the 100 randomly selected cases, how was the concordance between the 2 experts? It seems like correcting the analysis only for the cases where the AI's decision is different from the expert #1 is biasing the correction.

Thank you for your comment. In the 100 randomly selected cases, 15 were positive by the first pathologist, and only one of them was negative by the second one. There were no negative cases by the first pathologist that were positive by the second one. Although the concordance was higher for the 100 random cases than the cases alerted by the system (low-PS positives), correcting the analysis only for the cases alerted by the system indeed biases the performance for BCCA-test. We added a remark regarding this in line 247.

Conclusions:

* " Finally, our system can be applied to whole 380 slide images by automatically selecting tiles from regions of interest" - Why not try on some images available from TCGA database? Please see the first answer to Reviewer 1.

* I like the " Data interpretation by feature space visualization" section. Any comment on whether those features make sense, or have any biological meaning re what is known of those mutations and type of cancer?

Thank you for your comment. Indeed, it appears as if the features found by the system are in line with recent findings regarding the correlation between PDL1 expression, TMB, and tumor subtypes. We added a paragraph in the data interpretation section discussing this in lines 307-316. To shed more light on the system's decision, we also performed several new statistical experiments and added them in the same section.

References: The manuscript seems to lack references and would benefit from being put more in perspective to other existing work.

For example, although done on a different cancer type, I don't see discussion re Puladi, Behrus, et al. "Automated PD-L1 Scoring Using Artificial Intelligence in Head and Neck Squamous Cell Carcinoma." *Cancers* 13.17 (2021): 4409

How does your approach differs from that one and why a different one would be needed for breast cancer?

Thank you for your comment. In this paper, the authors have developed an AI-based system for prediction of PD-L1 expression in Head and Neck Squamous Cell Carcinoma. Their method was based on analysis of PD-L1 immuno-stained images. Their focus is on enhancing PD-L1 expression quantification by automating the pathologists' interpretation, and they show performance that is comparable to inter-pathologist agreement. We added a reference to this paper in our introduction (line 67). In our study, we show that PD-L1 expression could be predicted from hematoxylin and eosin images, without using immunohistochemistry, and this is the big difference. Our approach exploits a new level of information which has the potential to bypass staining deficiencies and intratumor heterogeneity, on top of interpretation errors.

Other papers in the field seem to be ignored - A few examples:

Kapil, Ansh, et al. "Deep semi supervised generative learning for automated tumor proportion scoring on NSCLC tissue needle biopsies." *Scientific reports* 8.1 (2018): 1-10.

Widmaier, Moritz, et al. "Comparison of continuous measures across diagnostic PD-L1 assays in non-small cell lung cancer using automated image analysis." *Modern Pathology* 33.3 (2020): 380-390.

...

Also, authors may want to look at that recently published paper:

Jin, Weiqiu, and Qingquan Luo. "When artificial intelligence meets PD-1/PD-L1 inhibitors: Population screening, response prediction and efficacy evaluation." *Computers in Biology and Medicine* (2022): 105499

The first two suggested papers focus on prediction of PD-L1 in non-small cell lung carcinoma. Similar to the paper suggested above, these studies show the ability of AI to repeat pathologists' interpretation by analyzing **IHC-stained** images. The third paper is an interesting review paper on AI and PD-L1/PD-1 in population screening, response prediction and efficacy. It is less focused on image analysis or PD-L1 expression prediction. We added the first two suggested papers to the introduction (line 67).

Reviewer 3 (Remarks to the Authors, Answers)

This paper introduces a new artificial intelligence deep-learning framework for identifying PD-L1 status from hematoxylin and eosin-stained (H&E) tumor microarray images. The proposed model was validated on an external dataset to demonstrate generalizability. While the authors demonstrate for the first time the ability to identify PD-L1 status from H&E stained images, the overall technique is relatively straightforward given the previously published deep learning frameworks used to identify estrogen receptor, progesterone receptor and human epidermal growth factor receptor 2, etc. Understanding this fact, the authors seek to demonstrate the translational relevance of their work as a tool that serves to reduce the challenges of PD-L1 assessment and act as a clinical decision support tool for pathologists.

As deep learning models for the identification of biomarkers in H&E stained images continue to proliferate, the originality of this paper is derived from its proposed ability to serve as an immediately applicable decision support tool by showing its performance on cross validation and an external dataset. The authors focus on the generalizability of their model and argue that the model's ability to gain information from features within the TMA images allow it to have criteria for making its predictions that are less error prone than human assessment (ground truth). However, the authors base this conclusion on an exceedingly small proportion of their data—two cases in which the model disagreed with the first pathologist and agreed with the second pathologist. The significance of the model's agreement with only one of the pathologists is questionable considering that there were four cases in which both pathologists had the same agreement and the model disagreed. More robust computational explanations would be welcome.

The value and statistical power of the system could be primarily justified by its performance on the entire cohort, rather than on the few discordant cases. Applied to both the validation folds, BCCA-test, and MA31 test sets, we showed that the system obtained high AUC performance, and that cases classified as low-PS were almost certainly negative (NPV = 97-100%), which were all statistically significant results. According to these measures, discrepancies between the pathologists that are unrelated to the H&E images (for example, due to non-specific IHC staining that could occur independently of the tumor morphology), falling in the low-PS group, would be almost certainly discovered by the system and alerted for review. In fact, the probability of discovering a false positive misclassification that occurred randomly in our data, is the percentage of low-PS cases out of the negative cases (64.6% on BCCA-test and 74.7% on MA31). Thus, it is reasonable that a large part of the pathologists discrepancies would be discovered by the system, and that system alerts (which are low-PS but positive cases) are likely to detect those discrepancies. Note that even those cases that were low-PS by the system but positive by **both** pathologists, could actually be negative and misclassified by both pathologists (for example, this could occur when there is an overrestaining of the IHC staining).

Specifically, referring to the comment, in the MA31 data the first pathologist marked 23 cases as positive, and out of these cases, the second pathologist classified 4 as negative. The system gave an alert for 2 of the cases, i.e., classified 2 of the 23 positives as low-PS, and both cases were among the discordant ones. The probability that these 2 alerts would fall within the 4 discordant ones by random chance is low: $(3^4)/(22 \times 23) = 0.0237$.

Thus, although the number of discordant low-PS cases is low, it is still enough to convince that the system's alerts indeed detected discordant cases, not by random chance. The same analysis can be done for the second pathologist who marked 25 cases as positive. Out of these cases, the second pathologist classified 6 as negative, and the system classified 2 as low-PS. Again both of these 2 cases from the 6 discordant ones, and the P value in this case is $(5*6)/(24*25)=0.05$.

Please note that this does not mean that the system predicts which cases are likely to be prone to misclassification based on the image features, without knowing the pathologist's classification. In other words, we did not have to assume that discordant cases have unique features that could be detectable, in order to do the above analysis, or in order to claim that such a system is useful for quality assurance. It is rather possible to predict the potentially misclassified cases **given** a pathologist's classification. The easiest way to understand this point is to imagine there is a special morphological feature (for example, tumor cells with a special shape) that only exists in negative cases. Let us assume there exists an AI system that classifies cases to low-PS when it detects this feature in certainty (and assume a certain detection is always correct). In this case, 100% of the system alerts (low-PS who were classified positive by pathologists) would be misclassifications of pathologists, even if the misclassifications were selected at random from the data, unrelated to any detectable feature.

Finally, to show that we did not have luck in the configuration of discordant cases between the first and second pathologists, let us assume we had another third pathologist. Assume the third pathologist's classification was also discordant from the first pathologist, but not necessarily in the same places. To maintain the same inter-pathologist variability rate, we randomly selected 10 cases (4 negatives and 6 positives by the first pathologist), made them the discordant ones, and repeated the analysis. Then, we measured the percent of system alerts (low-PS & positive by the first pathologist) that were indeed discordant between the first and third pathologist. We repeated this experiment 10,000 times. As a result, we obtained that the probability that a system alert would detect a discordant case is 0.992 (95% CI: 0.857 – 0.100).

Specific Comments:

1. (95) "This cohort consists of 652 recruited patients with ERBB2-positive metastatic breast cancer"

The authors assume that model validation on this cohort of patients remains valid despite the population consisting of only patients with ERBB2-positive, whereas the training data consists of all breast cancers. While this is perhaps a valid assumption, authors should address this in their analysis or discussion. For example, is there an association between ERBB2 positive and PDL1 positive in the validation set? Or between triple positive and PDL1 positive in the training set?

Please see our combined answer to this remark and remark #4.

2. (190) "NPV = 0.989 and a specificity of 0.913"

This is somewhat confusing, as specificity is not reported in Table 2. Authors should also compare specificity.

Thank you for noticing. It was a typo and should have been written “sensitivity”. We fixed it in the text and as requested, we reported the specificity in Table 2 for all experiments. Please also note the interpretation for specificity we added in the caption.

3. (242) “On the MA31 test set, both the NPV and sensitivity improved to 1.0 (perfect score), showing that any patient classified by the system as low-PS and positive by the first pathologist, was interpreted as negative by the second pathologist, in favor of the system (Figure 3b, MA31). This implies that the system’s predictions from H&E images alone may reveal additional information that could guide decisions in clinical practice.”

The authors might soften this claim given the low number of applicable patients.

In light of our explanations above, please let us know if you still wish for us to add any softening remark in the paper, or the P value calculation we showed above.

4. (277) “One can see that patients with low and high prediction scores were grouped almost separately, indicating that they have distinctive tissue characteristics.”

This analysis is beneficial and adds to the explainability of the model. However, are there other factors (triple-positive, date of diagnosis, etc) which explain this distribution? Additional efforts to demonstrate the model’s explainability could serve to provide additional justification for the previous claims that the model can have superior robustness to cases where there is pathologist disagreement. What are the features that lead to disagreement between the model and the pathologist, or between pathologists? Can these features themselves be used to identify images that should undergo additional pathological review or be restained?

Thank you for your proposals here and in comment #1. To better understand the system’s prediction in each subtype and how it could affect the generalization of the system to the MA31 data, we performed an analysis per tumor subtype. For each subtype, we plotted the t-SNE distribution and measured the AUC:

The t-NSE analysis showed visually that both high-PS and low-PS groups contained all possible subtypes, and that none of the subtypes alone, including ERBB2+ and Triple positive (Luminal ERBB2+), explained the system prediction score distribution. This was in line with the previously shown ability of the system to generalize to the MA31 cohort.

Interestingly, the ERBB2 enriched (ER-, PR-, ERBB2+) contained almost only high-PS scores, which could mean that the image features indicating PD-L1 negativity in high certainty (low-PS group) may not be found in the ERBB2 enriched group. This, however, does not mean there are only/mostly PD-L1 positives in this group (see #patients in the bar plot). Also, this finding should be softened given the low number of cases in that group. In any case, it would have been more useful if one of the subtypes contained mostly low-PS cases, rather than mostly high-PS cases.

The AUC analysis showed that there were no drastic changes in the performance when constraining to the specific subtypes, and that all AUC results were in the 95% CI of the total AUC. To shed more light on the system's decision, we also performed univariate and multivariate analysis, which showed that the expression of various molecular biomarkers added only a small amount of information that could contribute to the PD-L1 prediction, on top of the AI features information. We added the new analysis and interpretation in the "Data interpretation by feature space visualization" section, as well Supplementary Figure 4, and Supplementary Table 1.

The cases that had disagreement between the pathologists (other than those stemming from random misclassifications) are likely to be those with unclear PD-L1 expression (by IHC TMAs and not H&E) or unclear tumor (by either IHC or H&E). PD-L1 status may be unclear due to, for example, nonspecific IHC staining or PD-L1 expression close to 1%. The features that lead to disagreement between the pathologists and the system are those belonging to cases for which the system was confident (low-PS) but failed to predict the "ground truth" annotation. In principle, if these features had characteristics identifiable by the system (and probably by humans), the system would have been able to identify them and further increase its prediction performance. The multivariate analysis did not contribute to the prediction of the discordant cases. Thus, these features could probably not be explained by our data.

Again, the system does not predict which cases would be discordant, just by the image features. It rather predicts the PD-L1 expression by H&E, and then alerts on the error-prone cases based on discrepancies between its prediction and the pathologist's quantification by IHC. In other words, one can make use of the morphological information (H&E) for enhancing the PD-L1 prediction. IHC staining may fail or be unclear, while morphology, seen in H&E, remains unaffected. In such cases, the morphology information may provide a safety net, and the system could alert when there is a significant discrepancy between the IHC interpretation by pathologists and the H&E analysis by AI.

5. (322) "The calibration required held-out samples from MA31 that were then excluded from the final test and showed that indeed performance was increased (Figure 2b)."

This was somewhat confusing? What is the value of calibration when the testing set is then reduced in size—won't reducing the size of the test set and essentially increasing the number of training images always lead to increased fit?

Thank you for your comment. The calibration we used follows the "transfer learning" methodology which is a common practice procedure when transferring a model to a new cohort. According to this approach, a calibration step is done by fine-tuning the parameters of the last layer of the deep network using samples from the new cohort. Since the last layer has few parameters with respect to the rest of the network, a relatively small number of

examples from the new cohort are sufficient for calibrating the network to the nature of the new cohort and for increasing the performance, and the calibration is relatively fast. Transfer learning helps to improve performance primarily because the network is better fitted to the distribution of the new data, and not because more training samples were added (see Torrey, Lisa, and Jude Shavlik. "Transfer learning." Handbook of research on machine learning applications and trends: algorithms, methods, and techniques).

We designed a small experiment in order to show this point on our data. Instead of using samples from the MA31 data for the calibration step, we used the samples from the BCCA-test set for the calibration (860 patients), and then applied the system to MA31. Note, that for this experiment it is of course valid to use the BCCA-test for training because the system is then tested on another independent test set. As a result, the AUC on MA31 after this calibration was 0.854 (95% CI: 0.771 - 0.903), which means it remained completely unchanged (0.854 before the calibration), even though the BCCA-test set enlarged the training data by 860 patients, which is much more than the MA31 samples (~130 patients) used for the calibration.

6. Given the imbalance in the data, Precision-recall should be reported in parallel with ROC throughout.

Thank you for your suggestion. In our data, precision-recall (or PPV-sensitivity) curves may be very noisy due to the division in a small number of positive cases.

We suggest instead the NPV-specificity curves which are the exact opposite in terms of switching negative and positive. They hold the same balancing information but are not as noisy.

We added the NPV-specificity curves as Supplementary Figure 2.

Reviewers' Comments:

Reviewer #1:

Remarks to the Author:

The authors mostly solved my concerns. It is interesting to see that GradCAM is not valuable in understanding the models. Therefore, please add the discussion of the GradCAM part into discussions and supplemental figures.

Reviewer #2:

Remarks to the Author:

Following the authors' responses, I have 3 comments:

- Regarding the code and github pages:

The readme file for "https://github.com/amirlivne/PD-L1_predictor" is not informative enough for anyone to be able to use the code. Please follow usual code publishing guidelines.

- Regarding the TCGA tests: the authors say the slides "contain[] not only a tumor but also stroma, background" which can bias the results. The TCGA stores information regarding the tumor content/purity. If you check the AUC using only the slides where the tumor content is high, you will less likely be biased by those extra features and get something closer to TMA.

- Regarding the optimization of the parameters – The authors answer: "In the "Methods" section, we described our model architecture which defines the parameters that are optimized."

Sorry but, I don't see any more details regarding the hyper-parameters. I can't find mentions to batch size, learning rate, etc... Also, I think the Focal Loss function has some parameters to be optimized (alpha, gamma), and I don't see what was tested or optimized in that regard.

Reviewer #3:

Remarks to the Author:

Thanks to the authors for their revisions. In general the paper is growing stronger with these edits. I feel like a number of specific issues would still benefit from specific attention:

1) In my original comments, I asked for clarification regarding the discordance analysis as our first point of comment, ending with "The significance of the model's agreement with only one of the pathologists is questionable considering that there were four cases in which both pathologists had the same agreement and the model disagreed. More robust computational explanations would be welcome." Follow-up comments are provided here:

- I do not see where these explanations were provided in the edited text.

- I am not totally certain how the "p-values" provided in the rebuttal have been calculated.

Perhaps I am uncertain on the actual calculations, but these appear to be probability estimates of the likelihood of the event, not the p-value which is the probability of an event as or more extreme than the observed event.

2) The new tSNE based analysis for figure 4 is helpful and reinforces the robustness of the model. Please provide tSNE axis labels and color legend for figure 4a. However, it would still be beneficial to specifically address the question of what features are present in the discordant images that may be difficult for pathologists to interpret. Was there a significant difference in the subtype composition of discordant images compared to non-discordant images? Are there any identifiable morphological features in the discordant images?

3) The rationale for requesting AUPRC values was so that we can compare them to the underlying class proportions. In relatively imbalanced data, the AUROC can be misleading. Given the

imbalance in the data, please provide AUPRC values in addition to AUROC and class prevalence data.

To all reviewers

Please find below a point-by-point response to all of your comments referring to the relevant changes made in the manuscript. We also attach a copy of the **second** revised manuscript with all modifications in color for your convenience.

Reviewer #1 (Remarks to the Authors, Answers)

The authors mostly solved my concerns. It is interesting to see that GradCAM is not valuable in understanding the models. Therefore, please add the discussion of the GradCAM part into discussions and supplemental figures.

Thank you for your review. We added the GradCAM analysis to the supplementary material under the section “The Grad-CAM method for better understanding the system’s decision making”, accompanied by Supplementary Figure 6. We referred to this section in the discussion part in line 427.

Reviewer #2 (Remarks to the Authors, Answers)

Following the authors' responses, I have 3 comments:

- Regarding the code and github pages:

The readme file for “https://github.com/amirlivne/PD-L1_predictor”; is not informative enough for anyone to be able to use the code. Please follow usual code publishing guidelines.

Thank you for your feedback. We updated the readme file according to the guidelines.

- Regarding the TCGA tests: the authors say the slides “contain[] not only a tumor but also stroma, background” which can bias the results. The TCGA stores information regarding the tumor content/purity. If you check the AUC using only the slides where the tumor content is high, you will less likely be biased by those extra features and get something closer to TMA.

Thank you for your comment. This is an interesting suggestion. We searched for such data and contacted the GDC support as well. We found “tumor percent” and “nuclei percent” data but this data is given globally (per slide), and sometimes per region (top/bottom/etc). To screen out patches with no tumor, we would need higher resolution maps of the tumor location. If you are aware of such high resolution maps, we would be happy to know where we can find them, regardless of the manuscript.

- Regarding the optimization of the parameters – The authors answer: “In the “Methods” section, we described our model architecture which defines the parameters that are optimized.”

Sorry but, I don’t see any more details regarding the hyper-parameters. I can’t find mentions to batch size, learning rate, etc... Also, I think the Focal Loss function has some parameters to be optimized (alpha, gamma), and I don’t see what was tested or optimized in that regard.

Thank you for your comment. It is now more clear to us what you meant by explaining what are the parameters used. We added a description of the non-default hyper-parameters to the Methods section in line 527.

Reviewer #3 (Remarks to the Authors, Answers)

Thanks to the authors for their revisions. In general the paper is growing stronger with these edits. I feel like a number of specific issues would still benefit from specific attention:

1) In my original comments, I asked for clarification regarding the discordance analysis as our first point of comment, ending with “The significance of the model’s agreement with only one of the pathologists is questionable considering that there were four cases in which both pathologists had the same agreement and the model disagreed. More robust computational explanations would be welcome.” Follow-up comments are provided here:

- I do not see where these explanations were provided in the edited text.

- I am not totally certain how the “p-values” provided in the rebuttal have been calculated. Perhaps I am uncertain on the actual calculations, but these appear to be probability estimates of the likelihood of the event, not the p-value which is the probability of an event as or more extreme than the observed event.

The P value provided in the rebuttal was calculated as the probability of the event (“both system alerts were discordant”) because this event was already the most extreme one. The system did not alert any of the 19 concordant cases out of the 23, which is equivalent to obtaining sensitivity=100% (or NPV=100%). The P value in this case, can be calculated as the probability for obtaining sensitivity/NPV of at least 100% with a random classifier, which is the same calculation presented in the point-to-point response ($P = 0.0237$).

Because it is important that the system alerts would be indicative of discordant cases, the sensitivity or NPV are appropriate measures in this case. Nevertheless, one could alternatively calculate the P value of other measures. For example, the balanced accuracy (BACC), which is calculated as the average of sensitivity and specificity, could also be an appropriate measure when the data is highly imbalanced (prevalence of 4/23 discordant cases), and could be used instead. In this case, the BACC rate for the success of the alerts to detect the discordant cases is 0.75, since sensitivity=1 and specificity=0.5. The P value of a random classifier for obtaining BACC of at least 0.75 on the same data is 0.02, showing again that the alerts are likely to be correlated to the discordant cases.

Following this remark, we added a few modifications and comments along the paper regarding the likelihood of the alerts to be indicative, and regarding the probability of a random false positive miss-classification to be alerted (lines 221, 235, 413).

2) The new tSNE based analysis for figure 4 is helpful and reinforces the robustness of the model. Please provide tSNE axis labels and color legend for figure 4a. However, it would still be beneficial to specifically address the question of what features are present in the discordant images that may be difficult for pathologists to interpret. Was there a significant difference in the subtype composition of discordant images compared to non-discordant images? Are there any identifiable morphological features in the discordant images?

Thank you for your suggestions. We added the axes and legend to Supplementary Figure 4a. The questions about the discordant cases features are indeed interesting, but please note that: (1) The pathologists base their annotations on the IHC images. Thus, the discordant cases are likely to relate to unclear IHC staining and not to specific tumor subtypes or features in the H&E images that are difficult or not for the pathologists to interpret. 2) In our study, we did not investigate why or which subtypes are more difficult for

pathologists to annotate for PDL1, but rather assume that noise exists in the annotation process, and develop a system that may provide alternative/additional information for the classification based on H&E images. 3) The proposed system does not detect features that are more difficult for pathologists to interpret, but rather detects features that predict the PDL1 status, and the former cannot be concluded from the latter. Following the reviewer's suggestions, to see if any differences could be noted between discordant and non-discordant cases, we checked the subtype of each discordant case in the BCCA-test cohort. To visualize the subtype distribution of the discordant cases, we marked them on top of the tSNE plots:

This visualization demonstrates that the discordant cases were spread among the subtypes and there was no notable difference in the subtype composition of discordant images compared to non-discordant ones. We then showed the expert pathologists the H&E images of the MA31 discordant cases, and for comparison, other randomly selected H&E images from the non-discordant ones.

*Dashed dark framed images - H&E images of the 10 discordant cases from MA31 cohort.
Gray framed images - randomly selected H&E images of the non-discordant cases from the MA31 cohort.*

The pathologists reported that there were not any evident differences in the image features or tissue architecture between those two groups.

This analysis shows that even though the discordant and non-discordant cases may have indistinguishable H&E features, it could still be possible for a system to detect potential misclassifications; by predicting the PD-L1 status and comparing the prediction to the pathologist. Also, this shows that misclassifications may occur for both low-PS and high-PS groups (which do have different features). We added the discordant cases subtype distribution visualization to Supplementary Figure 4a and the MA31 discordant cases figure as Supplementary Figure 5, and we refer to them in line 336.

3) The rationale for requesting AUPRC values was so that we can compare them to the underlying class proportions. In relatively imbalanced data, the AUROC can be misleading. Given the imbalance in the data, please provide AUPRC values in addition to AUROC and class prevalence data.

Thank you for your clarification. We now better understand the rationale for the requested plots. For each plot, we now note the area under the curve (AUC) and the baseline (the prevalence of the corresponding class in the data) for comparison.

We added the precision-recall and NPV-specificity plots as Supplementary Figure 2.

Reviewers' Comments:

Reviewer #2:

Remarks to the Author:

"- Regarding the TCGA tests: the authors say the slides "contain[]" not only a tumor but also stroma, background" which can bias the results. The TCGA stores information regarding the tumor content/purity. If you check the AUC using only the slides where the tumor content is high, you will less likely be biased by those extra features and get something closer to TMA.

Thank you for your comment. This is an interesting suggestion. We searched for such data and contacted the GDC support as well. We found "tumor percent" and "nuclei percent" data but this data is given globally (per slide), and sometimes per region (top/bottom/etc). To screen out patches with no tumor, we would need higher resolution maps of the tumor location. If you are aware of such high resolution maps, we would be happy to know where we can find them, regardless of the manuscript."

I didn't mean select the tiles or sub-region, but I meant select the whole TCGA slides which "tumor percent" is high enough. For example, if the tumor percent is above a certain threshold (say 85% or something like this), it is safe to assume that most of the slide is covered with tumor. Therefore, that subset of slides could well be used as such without need to screen out patches.

Other concerns have been addressed.

Reviewer #3:

Remarks to the Author:

The authors have resolved my inquiries, and their additions are helpful in understanding the performance of the model and the practical future applications.

We thank the reviewers for their thorough review. Please find attached our final version of the manuscript (only formatting changes were done).

Reviewer #2 (Remarks to the Authors, Answers)

"- Regarding the TCGA tests: the authors say the slides "contain[] not only a tumor but also stroma, background" which can bias the results. The TCGA stores information regarding the tumor content/purity. If you check the AUC using only the slides where the tumor content is high, you will less likely be biased by those extra features and get something closer to TMA.

Thank you for your comment. This is an interesting suggestion. We searched for such data and contacted the GDC support as well. We found "tumor percent" and "nuclei percent" data but this data is given globally (per slide), and sometimes per region (top/bottom/etc). To screen out patches with no tumor, we would need higher resolution maps of the tumor location. If you are aware of such high resolution maps, we would be happy to know where we can find them, regardless of the manuscript."

I didn't mean select the tiles or sub-region, but I meant select the whole TCGA slides which "tumor percent" is high enough. For example, if the tumor percent is above a certain threshold (say 85% or something like this), it is safe to assume that most of the slide is covered with tumor. Therefore, that subset of slides could well be used as such without need to screen out patches.

Other concerns have been addressed.

Thank you for your clarification and suggestion. We tried this idea and indeed the results were better when using only slides with high tumor percent. We summarized the results in the table below.

Tumor percent	# Patients	# PD-L1+ Patients	AUC
≥ 0%	138	24 (17.4%)	0.683
≥ 80%	66	11 (16.6%)	0.719
≥ 85%	41	5 (12.2%)	0.817